

# Data-driven surrogate model for wind turbine damage equivalent load

Rad Haghi[1] and Curran Crawford[1]

[1]Department of Mechanical Engineering, Institute for Integrated Energy Systems, University of Victoria, British Columbia, Canada

**Correspondence:** Rad Haghi (rhaghi@uvic.ca)

**Abstract.** Aeroelastic simulations are used to assess wind turbines in accordance with IEC standards in the time domain. Doing so can calculate fatigue and extreme loads on the wind turbine's components. These simulations are conducted for several reasons, such as reducing safety margins in wind turbine component design by covering a wide range of uncertainties in wind and wave conditions, and meeting the requirements of the digital twin, which needs a thorough set of simulations for calibration. Thus, it's essential to develop computationally efficient yet accurate models that can replace costly aeroelastic simulations and data processing. We suggest a data-driven approach to build surrogate models for the Damage Equivalent Load (DEL) based on aeroelastic simulation outputs to tackle this challenge. Our method provides a quick and efficient way to calculate DEL using wind input signals without the need of time-consuming aeroelastic simulations. Our study will focus on utilizing a sequential Machine Learning (ML) method to map wind speed time series to DEL. Furthermore, we demonstrate the versatility of the developed and trained surrogate models by testing them for a wind turbine in the wake and using transfer learning to enhance their prediction.

## 1 Introduction

For design, optimization and maintenance purposes of a wind turbine, wind turbine researchers and engineers need to simulate a wind turbine's dynamic behaviour. This has been done based on the IEC standards (IEC 61400-1) using time-marching aeroelastic codes such as Fatigue, Aerodynamics, Structures, and Turbulence (FAST) (Jonkman et al., 2005), HAWC (Larsen and Hansen, 2007) or Bladed (Bossanyi, 2003). We utilize these time-marching simulations to calculate extreme and fatigue loads on wind turbine components caused by wind and wave as the inputs. The time-marching simulations are necessary for our work and research as they enable us to consider the inherent and necessary non-linearity in the wind turbine models. As both wind and waves are stochastic processes, a large set of simulations is preferred to understand the turbine behaviour fully and consider the uncertainty the stochasticity introduces. However, this increase in the number of simulations increases the computational costs. One solution to this is developing a computationally efficient Surrogate Model (SM) which is cheaper to run yet accurate for our purposes.



The concept of the SM can be traced back to the field of Uncertainity Quantification (UQ) analysis, as stated in Sudret (2007). SMs, emulators or response surfaces are simple representations of a complex model, which can map the input to the output. At the same time, they can encapsulate the complexity of the original model (Williams and Cremaschi, 2019). Asher et al. provides an overview of the different categories of SMs (Asher et al., 2015). There are different methods to develop a SM such as Polynomial Chaos Expansion (PCE) (Xiu and Karniadakis, 2002; Crestaux et al., 2009), or Gaussian Process Regression (GPR) (O'Hagan, 1978; Rasmussen and Williams, 2006). Recently, the application of Artificial Neural Network (ANN) and ML has become increasingly prevalent among researchers and engineers developing SMs (Wang et al., 2022; Kudela and Matousek, 2022; Dadras Eslamlou and Huang, 2022; Sun and Wang, 2019). This trend can be attributed to the widespread recognition of the ANN as a way to approximate any complex function with a few layers with high accuracy (Leshno et al., 1993), and the increase in the data accessibility and availability.

Researchers and engineers have been using SMs for increasingly diverse applications in the wind energy domain. In the load emulation domain, Dimitrov et al. (2018); Schröder et al. (2018); Dimitrov (2019) utilized PCE, Kriging, and ANN SMs to approximate wind turbine loads by considering stochastic variables such as turbulence intensity, mean wind, and wind direction. Avendaño-Valencia et al. (2021) employed a GPR-based SM to predict the fatigue load on a wind turbine affected by the wake in an onshore wind farm. Similarly, Shaler et al. (2022) used multiple SMs, such as GPR and ANN, to map inflow parameters in an array of wind turbines to the fatigue loads of the wind turbines in that array. Nispel et al. (2019) used a GPR-based SM for UQ of an offshore wind turbine's fatigue based on a wide range of environmental and structural variables. van den Bos et al. (2018) employed polynomial interpolation as an SM for estimating ultimate loads on a wind turbine, while Nielsen and Rohde (2022) used a random forest-based SM for ultimate load emulation. Singh et al. (2022) implemented a probabilistic SM for offshore wind turbine loads using chained GPR. Ransley et al. (2023) utilized an SM as an aerodynamic emulator for real-time testing of floating wind turbines. In a different approach, Fluck and Crawford used *intrusive* PCE to build a surrogate model for lifting line and Blade Element Momentum (BEM) models (Fluck, 2017; Fluck and Crawford, 2018). Similarly, Haghi and Crawford built SMs on a BEM model of NREL 5MW turbine simulations output time steps using *non-intrusive* PCE (Haghi and Crawford, 2022). In their work, the SMs mapped the random phases in the unsteady wind generation (Fluck and Crawford, 2018; Veers, 1988) to the output loads of the simulations at each time step.

As wind and waves are both uncertain, the high computational cost associated with the simulator in a Digital Twin (DT) may make it impractical to propagate uncertainty. Hence, employing a SM within the DT framework becomes beneficial when simulations are computationally expensive (Wright and Davidson, 2020). Also, using a surrogate model in a DT system creates the potential for the surrogate model to operate in real-time (Errandonea et al., 2020). In recent years, DT for wind turbines has gained popularity among researchers and engineers. DTs have been used at different levels in the energy systems and wind turbine industries. Song et al. provided an overview of DT applications and challenges for energy systems in the future (Song et al., 2023b). De Kooning et al. laid out an overview of DT applications in wind energy conversion (De Kooning et al., 2021). Fahim et al. provided a method to develop a DT for wind turbines in a wind farm-level system using machine learning methods





(Fahim et al., 2022). More specifically, with regard to DT applications for loads, Song et al. used measurements from the Block
Island offshore wind farm to develop a DT for the turbines in the field (Song et al., 2023a). In other work, Branlard et al. built
a DT based on a linearized model of a wind turbine (Branlard et al., 2020). Later, Branlard et al. developed a DT based on the
Tetra spar floating platform full-scale prototype successfully (Branlard et al., 2023). With numerous instances of successful
applications of DTs in the wind energy sector and the potential enhancements that a SM could bring to the DT framework, it is
crucial to conduct further research on developing accurate and efficient SMs for wind energy systems.

f

Recently, there has been a surge in using ML and ANN techniques to create wind system SMs. This subject has garnered
considerable attention and interest among professionals in the field. A recent study conducted by Schröder et al. utilized
Transfer Learning (TL) and physics-informed ML to enhance wind farm monitoring from Supervisory Control and Data Ac-
quisition (SCADA) data. The study aimed to improve the efficiency and effectiveness of wind farm monitoring using TL. The
results showed that integrating TL and physics-informed ML can enhance the accuracy and reliability of wind farm monitoring
systems (Schröder et al., 2022). Schröder et al. also used an ANN to build a SM that examined how changes in loads within
a wind farm affect the reliability of wind turbine components. Their study aimed to evaluate the impact of load changes on
wind turbine components' overall performance and reliability. The results showed that ANN-based SMs can provide valuable
insights into the behaviour of wind turbine components under different load conditions (Schröder et al., 2020). Additionally,
Mylonas et al. used a conditional variational autoencoder to create a probabilistic model of fatigue using SCADA data. Their
goal was to predict the probability of fatigue load in wind turbine components using SCADA data. The results showed that
ML-based methods predict fatigue accurately (Mylonas et al., 2021). Lastly, Dimitrov and Göçmen used a time-based ML
model Long Short-Term Memory (LSTM) to develop a virtual sensor that can predict and forecast the high-resolution load-
time series of wind turbine components based on a series of environmental and turbine behaviour variable inputs. The results
showed that ML-based time series models are accurate in their prediction and forecasting; however, a less complex ANN can
still effectively predict outcomes (Dimitrov and Göçmen, 2022).

## 1.1 Objective

The available literature and research indicate a lack of sufficient exploration and demonstration of a SM capable of mapping
high-resolution environmental time series, specifically wind and/or wave for both on- and off-shore wind turbines, to the fa-
tigue and extreme loads on wind turbine components. The development of such a SM could potentially enable the prediction
of the DEL of the wind turbine components using just a few input time series, thereby enhancing the efficiency of wind turbine
control systems and increasing the overall lifespan of the turbine. Moreover, the use of this system in a DT framework would
further enhance efficiency and facilitate real-time application.

Our ultimate is to develop a fully generalized SM that can predict wind turbine fatigue and extreme loads in any condition
without the need for extra customization or tweaking based on wind, wake, and wave time history. This manuscript specifically




begins to explore the approach by using sequential ML methods to build such a SM, which will map synthetic wind and wake time series to DEL. The objectives of the present manuscript are as follows:

- Building extensive wind time histories and wind turbine loads databases based on a Quasi Monte Carlo (QMC) sampling of the synthetic wind generation input variables.

- Developing simple Fully Connected Neural Network (FCNN) base SMs (Goodfellow et al., 2016) that maps synthetic wind generation inputs to DEL (Dimitrov, 2019), serving as a literature benchmark for performance and accuracy.

- Developing a sequential ML base SM using Temporal Convolutional Network (TCN) (Bai et al., 2018) to project syn-
100 thetic unsteady wind time series to wind turbine components DEL.

- Showing the capability of the *trained* sequential ML SMs by developing a TL framework to predict DEL while dealing with wake-induced synthetic wind time series.

### 1.2 Paper outline

This paper is organized as follows. Section 2 provides an overview of the methodology used in this study. The basics behind
the data-driven models are then described in Sections 2.1 and 2.2. Sections 2.3 and 2.4 explain the process of building the databases in detail. Section 3, we delve into the essential prerequisites for constructing the databases, imparting knowledge to the SMs, and leveraging their predictive prowess for both the free stream and downstream wake. In the same section, we also compare the accuracy of different SM architectures developed in this study and discuss the amount of data required for training, as well as the limitations of the developed SMs. The paper concludes in Section 4, where we summarize the main
findings of this work and suggest future research in the area of wind turbine surrogates using sequential ML models.

### 2 Methodology

The presentation of the methodology section in this document has been adapted from the approach outlined in Schröder et al. (2020) due to its clarity and relevance to the current topic. The chosen framework is deemed to be an appropriate and effective means of conveying the necessary information in a concise and organized manner. The methodology used in this manuscript
to map synthetic wind high-resolution time series to DEL is shown in Figure 1. It involves developing a sequential ML model combined with a FCNN architecture as the main SMs and utilizing a simpler FCNN for comparison purposes.

The configuration presented in Figure 1 has three blocks. The bottom block is for *Data generation*, which shows the procedure for building a database for the DEL from the input variables. The top two blocks are two methods to build a SM from the
120 generated data and input variables. The middle block presents the approach to building a SM that maps high-resolution wind time series to DEL based on TCN-FCNN architecture. The top block exhibits the process of creating a FCNN that projects the input variables to DEL (Dimitrov, 2019; Schröder et al., 2018). The larger frameworks and three blocks can be segmented into





twelve smaller stages. Each step is summarized below. Throughout this document, when we mention wind, we are specifically referring to unsteady wind.

(1) Specify the input variables space, their distributions and boundaries, and afterwards, generate $n$ samples $\mathbf{X}$ from the predefined variables. To enable tracking, every sample has been indexed. The database is split into two for training and testing:

(1a) *Training Input Variables* which includes $90\%$ of the samples randomly selected. Therefore, the size of this database is $0.9n$. The indices of the randomly selected samples $idx_{input}$ have been stored.

(1b) *Testing Input Variables* which includes the $10\%$ remaining of the samples. As a result, the size of this database is $0.1n$

(2) The $n$ generated samples are the input to a wind generator. Each sample from the input variable space generates one synthetic wind time series with the length of $t$ time steps.

(3) The $n$ synthetic wind time series are stored in *Wind Database*. The database size is $n \times t$ where $t$ is the number of time 135 steps in the time series. For training/testing purposes, this database is split into two parts:

(3a) *Training Wind Database*, which includes $90\%$ of the main synthetic wind time series database randomly selected. Consequently, the size of this database is $0.9n \times t$. The indices of the randomly selected samples $idx_{wind}$ have been stored.

(3b) *Testing Wind Database*, which includes the remaining $10\%$ of the main synthetic wind time series database. The 140 size of this database is $0.1n \times t$.

(4) The *Wind Turbine Model* is an input to the *Aero-servo-elastic Simulator*. The model comprises three modules: aerodynamic, controller, and aeroelastic.

(5) *Aero-servo-elastic Simulator* is a time-marching solver that takes synthetic wind time series and wind turbine model as the input and delivers forces and moments, *loads*, time series at $l$ wind turbine components as the output.

(6) All the $n$ outputs of the previous step simulations are stored in a database. In the *Simulation Database*, each simulation includes the $l$ wind turbine components load time series for $t$ time steps for one sample from the input variable space. Therefore, the database size is $n \times m \times t$

(7) The time-series output is analyzed to determine the DEL of the loads on the $l$ wind turbine components.

(8) For every wind turbine component in the *DEL database*, each simulation output yields a single DEL data point. There-150 fore, the database size is $n \times l$. Every row in the DEL database has an index that corresponds to the index of its input variable sample. As we train two SMs with the database, we split the database into training and testing databases twice. Thus, there appears to be an overlap between the testing and training databases. However, as we have utilized them to train and test two distinct SMs, we do not anticipate any issues arising from this situation.





(8a) *Training DEL Database* members are selected based on the $idx_{input}$ indices. Therefore, this database includes 90% of the DEL and the size is $0.9n \times l$

(8b) *Testing DEL Database* which includes the remaining 10% members of the DEL databases. Hence, this database size is $0.1n \times l$

(8c) *Training DEL Database* members are selected based on the $idx_{wind}$ indices. Therefore, this database includes 90% of the DEL, and the size is $0.9n \times l$. As mentioned before, there is an overlap between this database and the database in 8a.

(8d) *Testing DEL Database* which includes the remaining 10% members of the DEL databases. Correspondingly, this database size is $0.1n \times l$

(9) The SM with FCNN composition trains and validates using the databases in 1a as the input and 8a as the output.

(9a) For testing, the trained FCNN SM takes the database in 1b as the input and provides *FCNN Prediction DEL* as the output.

(10) The SM with TCN-FCNN architecture trains and validates using the databases in 3a as the input and 8c as the output.

(10a) For testing, the trained TCN-FCNN SM takes the database in 3b as the input and provides *TCN Prediction DEL* as the output.

(11) By comparing 9a with 8b, one can determine the accuracy of the of the trained FCNN SM.

(12) By comparing 10a with 8d, one can determine the accuracy of the of the trained TCN-FCNN SM.

The aim to build and train a simple FCNN SM is to compare the accuracy and performance of TCN-FCNN SM to it. The FCNN SM is not the ground truth in this piece of work; however, it has proven to provide acceptable accuracy for the similar input variable space (Dimitrov, 2019; Schröder et al., 2018).

After building and training the TCN-FCNN, we will show its versatility by examining the SMs with a synthetic wind including wake time series input. In other words, we test the SMs for a turbine in the downstream wake of another turbine. We developed smaller synthetic wind time series databases with wake effects, simulation outputs and their DEL. Also, we use TL to improve the SMs' performance over the wake.

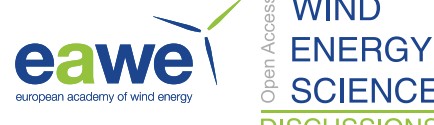

**Figure 1.** The data generation and SMs training and testing methodology



## 2.1 Fully Connected Neural Network Surrogate Model

After preparing the DEL database, we can begin training the SMs. The primary objective for the SMs is to map the input space to the output. Various mapping and regression methods are available for this task, but we suggest utilizing data-driven ML methods due to their ease of use and versatility. We developed two SM architectures; a FCNN and a TCN-FCNN. Here, the FCNN is a simple three-layer feed-forward ANN. The feed-forward ANNs are well studied and explained in the literature. For further explanation, we recommend referring to Goodfellow et al. (2016).

In order to train the FCNN, the input variable samples database is randomly divided into two parts: a training set comprising $90\%$ of the samples and a testing set comprising $10\%$ of the samples. These samples are uniquely indexed, and the training and testing sets indices are stored and tracked. The DEL database is similarly divided into training and testing sets, using the same indices as the input samples. To prevent data leakage, we ensure that there is no overlap between the training and testing databases. Once the training and testing databases are prepared, the FCNN is trained using the input variable space samples as input and DEL as output. The trained network is then tested using the testing input variables database to generate the prediction DEL. Finally, we compare the prediction with the testing DEL to measure accuracy. By following this process, we can ensure that the FCNN is accurately trained and tested, producing reliable results. Figure 2 shows the implemented network architecture. The input layer receives three input variables in the FCNN, while the output layer is responsible for the DEL. The weights on each neuron are determined through the training process using the weight optimizer. After the training, the FCNN is ready to predict the output based on the unseen (testing) data. Table 1 presents the FCNN model details.

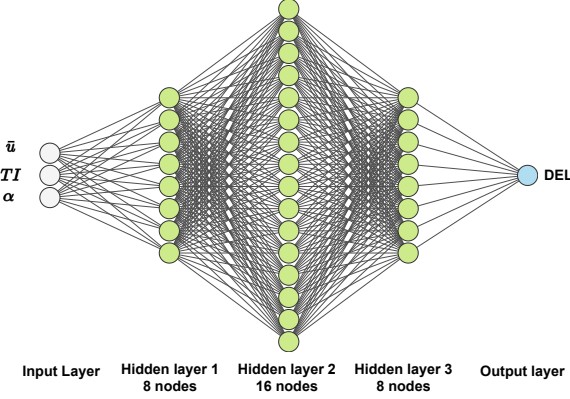

**Figure 2.** Architecture of the FCNN with three hidden layers. The number of nodes represents the implemented architecture

**Table 1.** FCNN architecture details

| Property | Value |
| --- | --- |
| Number of hidden layers | 3 |
| Number of nodes per layer | 8, 16, 8 |
| Number of trainable parameters | 321 |
| Activation function | ReLu |
| Learning rate | 0.001 |
| Cost function | Mean Square Error (MSE) |
| Training optimizer | Adam |





### 2.2 Temporal Convolutional Network-Fully Connected Neural Network Surrogate Model

In this section, we explain the TCN-FCNN architecture that we used to build a SM. Firstly, we provide an overview of the key components that make up a TCN. We will demonstrate how it can be effectively combined with a FCNN.

TCN is a novel approach that utilizes the benefits of a one-dimensional convolutional neural network to perform *sequential modelling* (Bai et al., 2018). One can define sequential modelling as a tool to map a sequential input $x_0, x_1, x_2, \ldots x_n$ to a sequential output $y_0, y_1, y_2, \ldots y_n$ as shown in Equation 1.

$$\hat{y_0}, \ldots, \hat{y_n} = f(x_0, \ldots, \hat{x_n}) \tag{1}$$

TCN is a member of the Convolutional Neural Network (CNN) family. CNNs have been used and are well known for classification proposes (Long et al., 2015). CNNs basics are well studied in the literature, and the interested reader is referred to Goodfellow et al. (2016); Long et al. (2015). Research has shown that TCN is better than Recurrent Neural Network (RNN) and LSTM in terms of performance, implementation, flexibility and versatility (Fawaz et al., 2019; Bai et al., 2018). TCN is based on three main concepts: a) the length of the output and input is the same, b) data should not leak from past to future. In

other words, the value of each data sequence in the output only depends on the past data sequences in the input, and c) it needs to be applicable to a long data sequence. To tackle these three, one can use the following techniques (Bai et al., 2018):

(a) The TCN employs a one-dimensional CNN architecture, wherein each hidden layer is of the same length as the input layer. To ensure consistent length, zero padding is incorporated in successive layers.

(b) In order to avoid data leakage, TCN utilizes *causal convolutions* architecture. In causal convolutions, the sequence $n$ of

215 the output solely relies on the sequences proceeding sequence $n$ in the prior layer.

(c) For the simple causal convolutions, the length of the sequences that it can capture is a multiplication of the network depth. It makes the model deep and computationally demanding for long sequential data with vanishing gradients. The solution to this challenge is to utilize the *dilated convolution*. By using dilated convolution, the network is able to increase its receptive field significantly in an exponential manner. For a one-dimensional sequential input $\mathbf{x}$, a filter $f$, and the

220 element $s$ of the sequence, one can define the operation $F$ as:

$$F(s) = (\mathbf{x} *_d f)(s) = \sum_{i=0}^{k-1} f(i) \cdot \mathbf{x}_{s-d \cdot i} \tag{2}$$

where $*$ is the convolution operator, $d$ is the dilation factor, $k$ is the kernel size and $s - d \cdot i$ points out the direction of the past. In dilated convolution, the dilation factor increases exponentially with the level of the network depth. Figure 3a shows an illustration of a dilated convolution. The history of the sequences that a layer can take into account is $(k-1)d$.





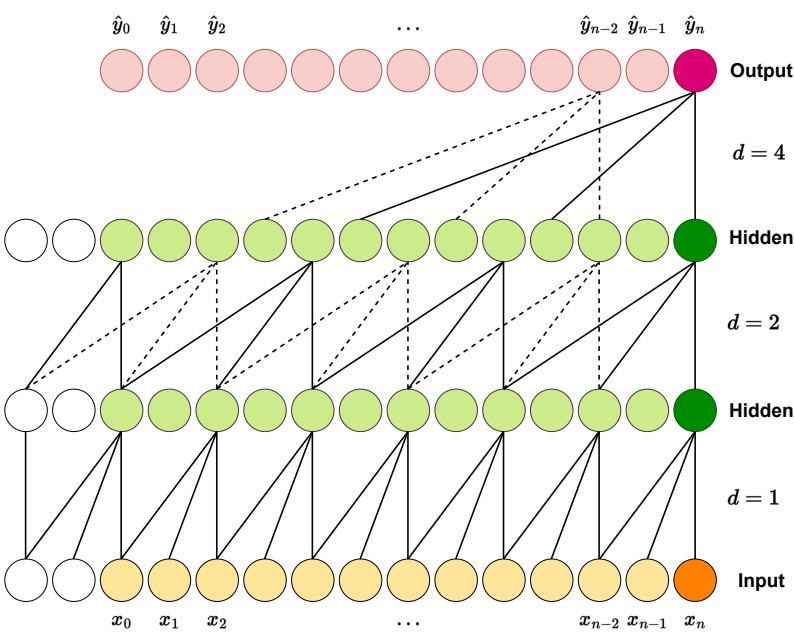

(a) Illustration for a dilated causal convolution example, with kernel size $k = 3$ and dilation factors $d = 1, 2, 4$. The receptive field has the ability to encompass all values present within the input sequence. The white circles show the zero padding in the layers.

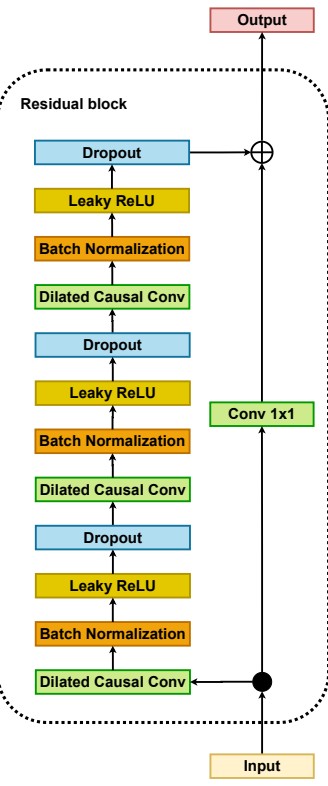

(b) TCN residual block used in this work

**Figure 3.** Dilated Causal CNN and the Residual Block for TCN

As the TCN network needs to take into account larger sequential data, it needs many layers and, as a result, gets deep quickly. This causes the network's problem of *performance degradation*, which needs to be stabilized. Therefore, we utilize a *residual block* as a replacement for a *convolutional layer* (Bai et al., 2018; He et al., 2015). The residual block methodology incorporates a branching mechanism where the input is injected into the output, passing through a CNN. The residual block used in this study is shown in Figure 3b.

     For this study, we utilized the aforementioned TCN to extract features from the input time series. *Feature learning* or *feature extraction* is the process by which the machine learning model converts the raw data into an "internal representation", feature vector or latent space (LeCun et al., 2015). Then this feature vector is employed to detect the output pattern through a secondary machine learning subsystem. In this study, we took advantage of TCN ability to extract features in the sequential data.

Thereafter, we used the features as the input to a FCNN. The integration of TCN and FCNN enabled us to map the wind time series into DEL. Westermann et al. used a similar approach but for a different application in Westermann et al. (2020). The ex-





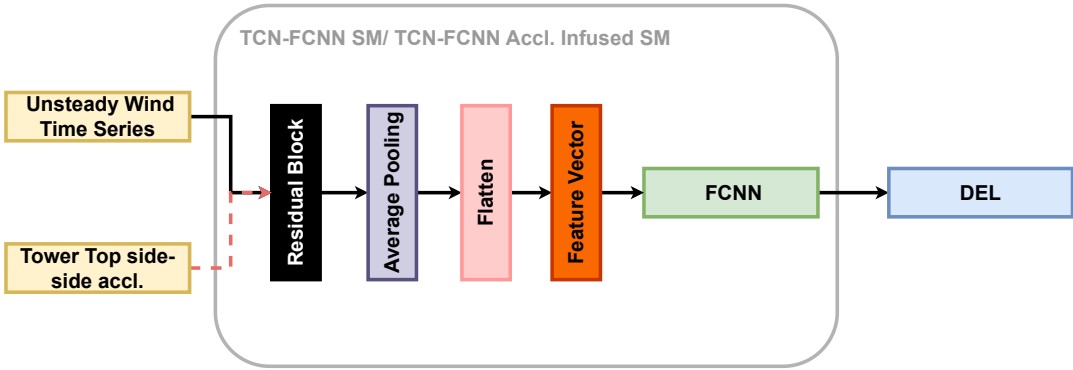

**Figure 4.** The TCN-FCNN architecture. The tower top side-side acceleration is an optional input that we will discuss further in Section 3.5

plained technique is illustrated in Figure 4. The "Residual Block" in Figure 4 made up of the components depicted in Figure 3b.

With all the requisite SM components in place, we can proceed with the training and testing phases. As mentioned before, we indexed the samples and TurbSim generated wind time series outputs. In the same manner as explained in Section 2.1, the synthetic wind time series database is divided into training and testing databases, where $90\%$ of the database is randomly selected for the training, and the remaining $10\%$ goes for testing. As the indices for the training and testing databases are known, they are used to divide the DEL database into training and testing databases. As the selection of the training and testing indices is random, the DEL training and testing databases members are different from the ones explained in section 2.1. With the training and testing databases ready, the TCN-FCNN is trained on the training data. Afterwards, we utilized the trained model to forecast DEL using the synthetic wind time series that was not included in the training database. The predicted DEL is then compared with the testing DEL to measure the accuracy of the mapping. The specifications of the TCN-FCNN SM employed have been detailed in Table 2.





**Table 2.** TCN-FCNN architecture properties and details for both the main approach and TL approach. TL FCNN learning rates are for the initial training and fine-tuning consequently.

| Property | Residual block | FCNN | TL FCNN |
|---|---|---|---|
| No of Conv1D filters | 6, 6, 6 | - | - |
| Kernel size | 20, 13, 8 | - | - |
| Dilation factor | 1, 2, 4 | - | - |
| Drop out rate | 0.05 | - | - |
| Average pooling size | 100 | - | - |
| Activation function | Leaky ReLU | Linear | Leaky ReLU |
| Trainable parameters | 1950 | 1681 | 500 |
| Number of hidden layers | - | 3 | 3 |
| Number of nodes per layer | - | 16, 32, 16 | 8,11,8 |
| Learning rate | 0.001 | | 0.001, 0.00001 |
| Cost function | MSE | | MSE |
| Training optimizer | Adam | | Adam |

## 2.3 Variable input space boundaries, distributions and sampling

For the data generation, selecting the appropriate input variable space, the boundaries for each variable, and their distributions is crucial. Depending on the problem at hand, different input variables might be needed. As for this study we only considered one onshore wind turbine, only the input variables that affect the wind generation are considered. These three variables are mean wind speed $\bar{u}$, turbulence intensity $TI$ and wind shear $\alpha$. Therefore, the input space $\Theta$ can be defined as:

$$\Theta = [\bar{u}, TI, \alpha] \tag{3}$$

The boundaries and distributions of our input variables help define the conditions for which our models are designed. It is important to note that wind speed is considered an independent variable, while the other two variables' boundaries and distributions depend on the wind speed. We have selected the variables, their boundaries and distributions to build the database based on research presented in Dimitrov (2019). Our use of QMC Sobol's sampling method (Sobol', 1967) allows for accurate sampling of the predefined joint distributions in a deterministic non-repetitive manner. In this study, Sobol's sampling method is preferred as it is consistent and computationally efficient (Kucherenko et al., 2015). Also, this sampling method is reproducible and provides better uniformity properties of the samples over the distributions (Renardy et al., 2021). In the following, when we refer to the sample, it means a vector of three elements of mean wind speed, turbulence intensity and wind shear.





## 2.4 Simulation and Damage Equivalent Load databases creation

The samples from the input variable space are the input to the synthetic wind generator. Each sample from the space provides one input to the generator. The output of the generator is a "full field" synthetic wind time series (Jonkman and Buhl Jr, 2006). The synthetic wind generation basics are explained in length in Veers (1988). In this study, we employed TurbSim to generate the synthetic wind fields (Jonkman and Buhl Jr, 2006). From each sample, the three input variables $[\bar{u}, TI, \alpha]$ are directly taken into the TurbSim input file and generate one synthetic wind time series using TurbSim. In order to guarantee that every time

series created is distinct, a unique *seed* number is assigned to each sample. The output of the wind generator can be defined as a function of the sample:

$$\mathbf{U}(t, y, z) = f(\Theta) \tag{4}$$

The output has *spatial* and *temporal* components $\mathbf{U}(t, y, z)$. The spatial component of the full field synthetic wind comes from the *grid points*, which are defined over the wind turbine rotor plane. The output of Turbsim provides one time series of

275 synthetic wind at each grid point in $x$, $y$ and $z$ directions, namely $u$, $v$ and $w$. These time series are correlated to each other, depending on mean wind speed and their distance from each other (Veers, 1988; Jonkman and Buhl Jr, 2006).

The full-field synthetic wind is the input to the aero-servo-elastic simulations. To run these simulations, next to the synthetic wind time series, the aerodynamic model, aeroelastic model and controller model are required too. This study used an

280 onshore model of the National Renewable Energy Lab (NREL) 5MW reference wind turbine (Jonkman et al., 2009). The wind turbine model includes aerodynamic, aeroelastic and controller submodules. To run the simulations, we used OpenFAST, the time marching aero-servo-elastic solver developed by NREL (Jonkman et al., 2022). OpenFAST's output includes both temporal and spatial dimensions, with loads provided from various wind turbine components located at different positions, such as blades, towers, and gearboxes. This spatial aspect is integral to understanding the full scope of the data. The simulations in this

study follow the IEC standards for power production Design Load Case (DLC) 1.2, as stated in IEC standards (IEC 61400-1).

Thus far, we have established a database that comprehensively incorporates all the simulation output time series data. Once we have that, the data is processed to obtain the simulation time length statistics and DEL for evaluating the loads and fatigue. DEL calculation is based on the Palmgren–Miner linear damage rule as explained well in Thomsen (1998) and Stiesdal (1992).

DEL can be formulated as:

$$DEL = \left( \frac{\Sigma n_i R_i^m}{n_{eq}} \right)^{1/m} \tag{5}$$

In the given context, $m$ represents the Wöhler slope, while $R_i$ and $n_i$ correspond to load ranges and the respective number of cycles. The output is obtained through rainflow counting of the load time series (Thomsen, 1998). $n_{eq}$ is the equivalent number





of load cycles which is usually the length of the simulation in $s$. The DEL database includes all the calculated DELs from every
simulation at its outputs.

## 2.5   Simplified Wake Model

Once the SMs are built and trained, we assess their versatility by testing them with a turbine in the wake. To proceed, we must create a new database that includes the synthetic wind and DEL with consideration given to wake effects.

The wake caused by a wind turbine has been studied extensively and is out of this manuscript's scope. Different methods and models exist to implement wakes in the aerodynamic simulation of a wind turbine (Sanderse et al., 2011; Göçmen et al., 2016). For the sake of simplicity and ease of implementation, we limit the study to a simplified wake definition, with the study turbine in the wake of one turbine only. The simplified wake includes a non-uniform wind speed deficit and an increase in turbulence intensity across the rotor. For implementing wakes in the synthetic wind time series, we used the method explained
in William et al. (2022).

For the velocity deficit caused by the wake over the rotor plane, we utilized the super-Gaussian deficit (Blondel and Cathelain, 2020). We used the formulation developed by Ishihara and Qian (2018) for the added turbulence intensity model. Also, the same as Bastankhah and Porté-Agel and Ishihara and Qian, we are assuming the linear expansion of the wake that occurs
downstream of a turbine. The following are the steps we took to implement the downstream simplified wake model:

(1) Using Sobol's sampling method, take $2^n$ samples from the input variables $[\bar{u}, TI, \alpha]$ as explained in Section 2.3.

(2) Knowing the turbine thrust coefficient $C_t$ at each wind speed, the ambient turbulence of the free stream and the distance between the turbines, one can calculate the downstream wake width based on the formulation in Ishihara and Qian (2018).

(3) With wake width calculated in the previous step, one can calculate the velocity deficit (William et al., 2022; Blondel and Cathelain, 2020) and added turbulence intensity (Ishihara and Qian, 2018).

(4) Both calculated velocity deficit and added turbulence intensity have spatial distribution over the rotor plane. We considered this distribution by modifying the mean wind speed and turbulence intensity of the samples for the first step. For the added turbulence intensity, the average over the rotor disk is added to the $TI$ value of the sample. For the deficit wind
velocity, *harmonic mean* over the rotor disk is deducted from the samples mean wind speed $\bar{u}$ (William et al., 2022).

(5) With the *modified Sobol's samples* in our possession, we used TurbSim to generate synthetic wind time series from each modified sample. Remember that these synthetic wind time series have modified turbulence intensity and reduced mean wind speed, but the Gaussian deficit has not yet been included.





(6) For the generated synthetic wind by TurbSim, for the mean wind speed of $\bar{u}$, the wind speed in time $t$ at each $(y, z)$ point
can be defined as:

$$U(t, y, z) = \bar{u}(y, z) + \tilde{u}(t, y, z) \tag{6}$$

where $\tilde{u}(t, y, z)$ is the zero mean turbulence. The inclusion of the velocity deficit caused by the wake in the generated
synthetic wind can be expressed as:

$$U^*(t, y, z) = \phi(y, z) + \tilde{u}(t, y, z) \tag{7}$$

where $U^*(t, y, z)$ is the modified wind field and $\phi(y, z)$ is the velocity deficit distribution over the rotor $yz$ plane, which
we calculated in 3.

We will use synthetic wind with a velocity deficit and added turbulence intensity to run OpenFast simulations containing the
simplified wake model and calculate the DEL as described in Section 2.4.

## 2.6   Transfer Learning

According to Goodfellow et al., transfer learning aims to utilize what has been learned in one context to improve the "*generalization*" in another context (Goodfellow et al., 2016). For this study, we use TL for the cases with wake to improve their
prediction.

After following the steps outlined at the beginning of this section, we obtain trained SMs that are able to accurately predict
the DEL of wind turbine components under freestream synthetic wind conditions. We then use these models to predict DEL
in the wake of a turbine. We implement TL to enhance our predictions by loading the trained SMs and freezing their weights,
making them untrainable. Then, we remove the FCNN part of the TCN-FCNN and replace it with a trainable FCNN. The new
FCNN we are using now has a simpler architecture compared to the one we previously used for training and testing on the free
stream data. Essentially, we now have a frozen weight (untrainable) TCN along with a trainable FCNN. As with the previous
training process, we utilize $90\%$ of the wake databases for training and $10\%$ for testing. The training has two steps; the first step
is to train the aforementioned combination of untrainable TCN and trainable FCNN to the desired accuracy, and then *fine-tune*
the TL model by unfreezing the TCN part weights and training it on the same data again but this time with a smaller learning
rate. The properties and details of the TL FCNN are shown in Table 2. Figure 5 illustrates the architecture used for the TL.





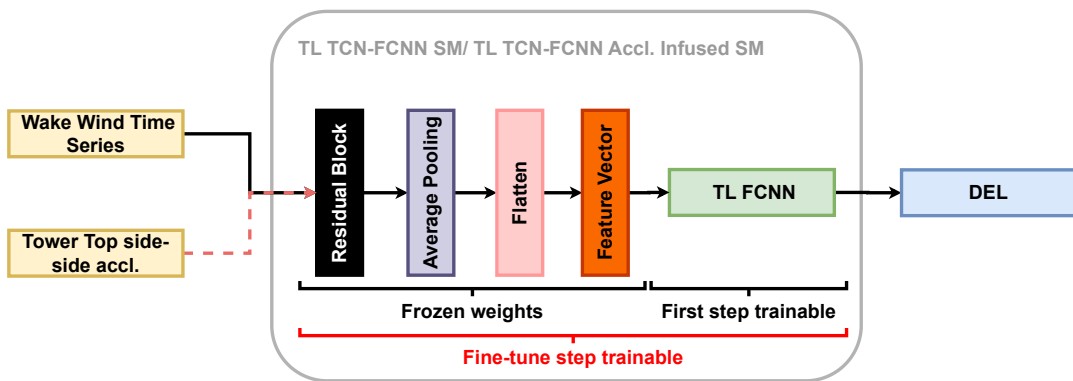

**Figure 5.** The frozen TCN with trainable simple FCNN for the first step of TL, the the fine-tuning step.

## 3 Results and discussion

To this point, we explained the background methods that have been used in this piece of work. This section will discuss the conditions that were utilized to generate the results.

### 3.1 Input variables boundaries, distributions and sampling

For generating the synthetic wind time series and DEL, we used the input variables space as it is explained in Section 2.3. The mean wind speed is sampled from a uniform distribution, where the boundaries are decided based on the NREL 5MW wind
turbine characteristics between the cut-in to cut-out wind speed (Jonkman et al., 2009). For every wind speed sample, we took a sample from the $TI$ and $\alpha$ too. The other two input variables are also defined as a uniform distribution, whose boundaries are a wind speed function. For the $TI$, the boundaries are based on the IEC class 1A values (IEC 61400-1). We choose the same boundaries for wind shear $\alpha$ as Dimitrov (2019). The input variables and their boundaries can be found in Table 3.

**Table 3.** The input variables boundaries

| Input variable | Lower boundary | Upper boundary |
|---|---|---|
| Mean wind speed $\bar{u}$ | $\bar{u} \geq 3m/s$ | $\bar{u} \leq 25m/s$ |
| Turbulence Intensity $TI$ | $TI \geq 0.04$ | $TI \leq I_{ref}(0.75 + 5.6/\bar{u})$ |
| Wind shear $\alpha$ | $\alpha \geq \alpha_{ref,LB} - 0.23\left(\frac{\bar{u}_{max}}{u}\right)\left(1 - \left(0.4\log\frac{R}{z}\right)^2\right)$ | $\alpha \leq \alpha_{ref,UB} + 0.4\left(\frac{R}{z}\right)\left(\frac{\bar{u}_{max}}{u}\right)$ |

where

- from IEA class 1A, $I_{ref} = 18\%$

- $\alpha_{ref,LB} = 0.15$ and $\alpha_{ref,UB} = 0.22$ are reference wind shear a 15 m/s

- $\bar{u}_{max} = 25m/s$ is the upper bound of the wind speed

- $R$ is the rotor radius, and $z$ is the hub height

Once we have established the joint distributions and boundaries, we can generate sample points for the input variable space. As described in Section 2.3, we used Sobol's sampling method for this study. The Sobol's samples need to be in the order of $2^n$, otherwise they lose their balance properties (Owen, 2021). Therefore, we took $n = 2^{15}$ samples from the predefined distributions. We decided to have a conservative number of samples, as Sobol's sampling method enables us to reduce the number of samples without losing the benefits of the method or resampling the domains. To generate an example of the variable space,

we took $2^{10} = 1024$ samples from the predefined distribution in Table 3. The samples and the input variable boundaries are displayed in Figure 6.

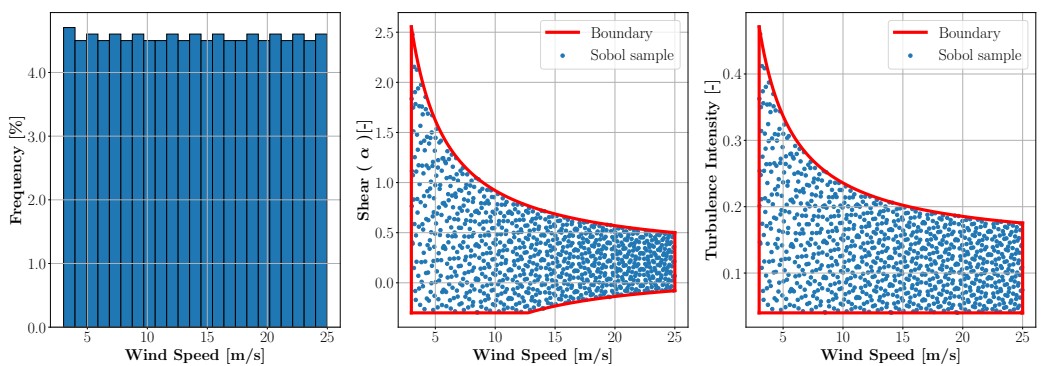

**Figure 6.** 1024 Sobol's samples from the predefined distributions for $\bar{u}$, $TI$ and $\alpha$ with the boundaries of the variables.

### 3.2 Turbsim and OpenFAST output

As mentioned before, we use each sample from the input variables to generate a unique synthetic wind time series using Turb-

370 Sim. Every TurbSim output generated from each sample is assigned a unique seed number to ensure that there are no repeating seeds. The TurbSim output format is well described in (Jonkman, 2009). For this study, we used a 15 by 15 grid over the rotor's plane. For training and testing purposes, we only took into account nine synthetic wind time series in $x$ direction out of 225 synthetic wind time series. Our tests show including the wind in $y$ and $z$ directions would not improve the training or testing results; therefore, they are omitted. These nine synthetic wind time series are approximately located at the middle of

375 the rotor and hub height. Our tests show other configurations of the points (e.g. circular layout of points) have little impact on the results. The grid points are selected to be roughly located at the blades' mid-span. This selection is illustrated in Figure 7a. Regarding the time component, the synthetic wind time series has a frequency of $20Hz$, with a duration of $720sec$. After running the simulation and later in the training/testing step, we upsample the synthetic wind to $1Hz$ due to the hardware constraint.

As mentioned before, we run aeroelastic simulations on an onshore NREL 5MW model using OpenFAST. OpenFAST can provide an extensive set of outputs, namely channels, at different components of the turbine. The channels and their descrip-



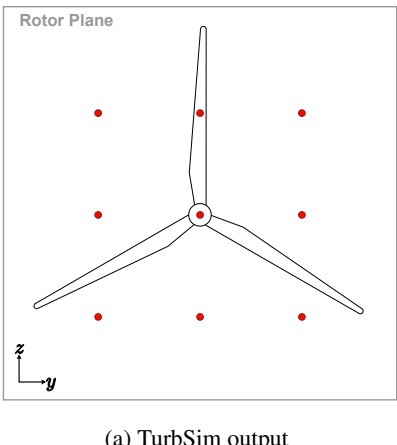
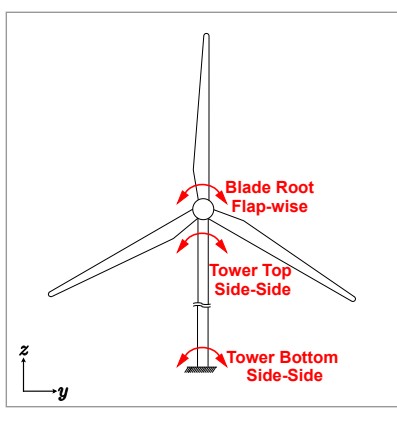
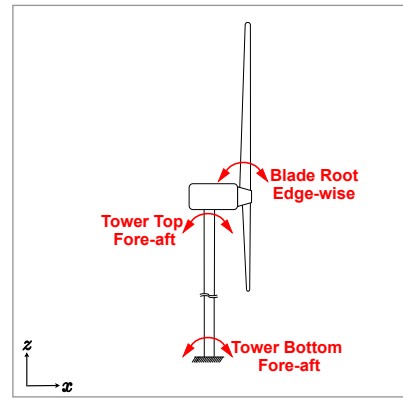

(a) TurbSim output     (b) OpenFAST output channels $y, z$ plane     (c) OpenFAST output channels $x, z$ plane

**Figure 7.** (a) Illustration for selected TurbSim output grid point locations as the input to the SM for training and testing. (b) and (c) show a schematic drawing of a turbine, with the output load channels.

tions can be found in Jonkman et al. (2005). For this study, we took into account six moment output channels and the average generated power for the training/testing objectives. These six moments are blade root edgewise and flap-wise moments, tower top fore-aft and side-side moments, and tower bottom fore-aft and side-side moments. Figure 7b and 7c illustrate a schematic

drawing of the wind turbine with the load channel locations that we used for training/testing in this study. The OpenFAST output channel label and its corresponding naming for this study are provided in Table 4.

**Table 4.** Channel label, naming and units

| OpenFAST Channel label | Naming | Post Processing | Unit |
| --- | --- | --- | --- |
| GenPwr | Generated Power | 10 minutes average | [kW] |
| RootMxb1 | Blade edgewise moment | DEL | [kNm] |
| RootMyb1 | Blade flapwise moment | DEL | [kNm] |
| YawBrMxp | Tower-top side-side | DEL | [kNm] |
| YawBrMyp | Tower-top fore-aft | DEL | [kNm] |
| TwrBsMxt | Tower bottom side-side | DEL | [kNm] |
| TwrBsMyt | Tower bottom fore-aft | DEL | [kNm] |

We run $n = 2^{15} = 32768$ aeroelastic simulations using OpenFAST for this study. We run the simulations in 2048 batches of 16 simulations in parallel using Digital Research Alliance of Canada resources. Each simulation ran for $720 sec$, but the first

$120 sec$ of the simulation output was discarded to avoid any initialization effect. The time step for the aeroelastic simulation was set to $0.00625 sec$, while the output resolution is $20 Hz$. After running all the simulations and building the simulations output database, we calculate the DEL for each simulation, for the interested output channels for Wöhler slope $m = 4$ and





$n_{eq} = 600$ in Equation (5). Also, we considered the 10-minute average of the generated power. To read the OpenFAST output files and calculate the DEL, we used python `pyfast` library (Branlard, 2023).

### 3.3 Turbine in wake output

Our aim is to test the effectiveness of the trained TCN-FCNN SMs by using wake input. Specifically, to determine if the model, which is trained on a turbine in the free stream, can accurately predict the DEL of a turbine in wake as well. The test scenario involves one turbine in free stream, which the SMs are trained on, and one turbine in downstream wake. The distance between the two turbines is $7D$, where $D$ represents the rotor diameter. In our specific case, we are considering the rotor diameter of $126m$ for NREL 5MW, which results in a distance of $882m$ between the two turbines.

We followed the same process described in Section 3.1 by taking $2048$ samples from the distributions outlined in Table 3. To calculate the wind velocity deficit and add turbulence, we used the Gaussian model (Blondel and Cathelain, 2020) and (Ishihara and Qian, 2018), respectively, as explained in Section 2.5. We then adjusted the $\bar{u}$ and $TI$ of each sample based on the harmonic mean of the wind velocity deficit and the arithmetic mean of the added turbulence intensity over the rotor. Using the modified samples, we generated 2048 Turbsim full-field outputs, following the process explained in Section 3.2. With the Turbsim output now available with modified $\bar{u}$ and $TI$, we utilized a Python script to offset the generated synthetic wind with the Gaussian velocity deficit profile (William et al., 2022).

In Section 2.5, it was explained that the wind velocity deficit has a distribution across the $yz$ plane. This distribution can shift across the rotor plane depending on the location of the wake centre. We established three wake cases with wake centres located at $(-30m, 90m)$, $(0m, 90m)$, and $(30m, 90m)$ on the rotor plane $yz$. The wake centre is assumed not to move vertically since both turbines have the same hub height of $90m$. Figure 8 illustrates an example of the velocity deficit effect on the TurbSim output. In Figure 8, the first column shows the TurbSim output with the added turbulence intensity, the middle column is the Gaussian velocity deficit for the aforementioned wake centers, and the last column is the first column with the velocity deficit offset. They are all a snapshot of the TurbSim output at $320s$, and the input samples are $[\bar{u} = 13.42m/s, TI = 11\%, \alpha = 0.107]$. The red circle is the rotor disk.

As all the $2048$ TurbSim outputs are at hand, one can run OpenFast simulations and calculate DEL as explained in Section 3.2.







(a) Wake centre at $(-30m, 90m)$

(b) Wake centre at $(0m, 90m)$

(c) Wake centre at $(30m, 90m)$

**Figure 8.** The velocity deficit implementation on a TurbSim output with the added turbulence intensity.

## 3.4 Training-Testing

Now that we have constructed all the necessary databases, we can begin the training process. We must first normalize the data before training the SMs. In this study, we employ *min-max scaling* for both input and output values, scaling them to a range





**Table 5.** Training setting for the SMs

| Parameter | TCN-FCNN | FCNN | TL TCN-FCNN |
|---|---|---|---|
| Batch size | 256 | 256 | 64 |
| Trainable parameters | 3631 | 321 | 500 |
| Maximum number of epochs | 3000 | 3000 | 3000 |
| Validation split | 5% | 5% | 5% |
| Early stopping - monitoring | validation loss | validation loss | validation loss |
| Early stopping - patience epochs | 300 | 300 | 3000 |
| Early stopping - best weight restoration | True | True | True |
| Input shape | (batch size, 600, 9) | (batch size, 3) | (batch size, 600, 9) |

of 0 to 1. The input variables are scaled individually, while all the synthetic wind time series, regardless of the mean wind
speed, are included in a single scaling procedure. During the training process of the TCN-FCNN and FCNN models, we imple-
mented a separate scaling of the DEL of each output channel. This approach was necessary to ensure the scaling was tailored
to the specific needs of each channel. A total of twelve SMs were trained, with six models being trained for each respective type.

The total number of samples in the dataset is 32768. As mentioned in Section 2.2, this dataset is split into two parts for
different purposes randomly by choosing the indices of the samples through a non-repetitive random number generator. The
training set contains $90\%$ of the samples, and the testing set contains $10\%$ of the samples. Rather than training the SMs on all
the training data, the training data set is divided into batches of 256 samples. Afterwards, the SMs trained on each batch, and
after going through all the batches of the training, one "epoch" is completed. The main optimization method used in ML com-
munity is Stochastic Gradient Descent (SGD) (Bottou, 2010). Taking all the training data set for the training while employing
SGD requires a large amount of memory, and SGD may land you at a "saddle point" (Ge et al., 2015). One can tackle both
of these issues in training by dividing the training data into batches, as explained before. One disadvantage of this method is
that it requires more epochs for the model weights to be fully trained and converged. In this study, we used Python package
`tensorflow` for the ML model development, training, and testing (Abadi et al.). The table outlining the settings and details
for training in `tensorflow` can be found in Table 5.

We employed early stopping for the training as it reduces the required training time. Once the training phase is complete, the
remaining $10\%$ of the data that was not used during training is utilized for testing purposes. The output of the testing procedure
provides the accuracy of the fitted models. For this study, we use *coefficient of detemination*, $R^2$, and Normalized Root Mean
Square Error (NRMSE) as the measures for the fitted models' accuracy.



## 3.5 TCN-FCNN results

This section provides the output of the testing process, as explained before, for the TCN-FCNN SMs. Figure 9 shows the results of the testing on the trained SMs for each output channel scaled DEL. Each plot in the figure delivers one channel, and data connects to the mean wind speed of the samples using a colour map. Upon reviewing the outcomes, it is evident that the SMs offer precise prediction based on synthetic wind time series data that it has not previously encountered. Based on the colour maps, it is inconclusive to determine the correlation between the input variables of the sample and the accuracy of the fit.

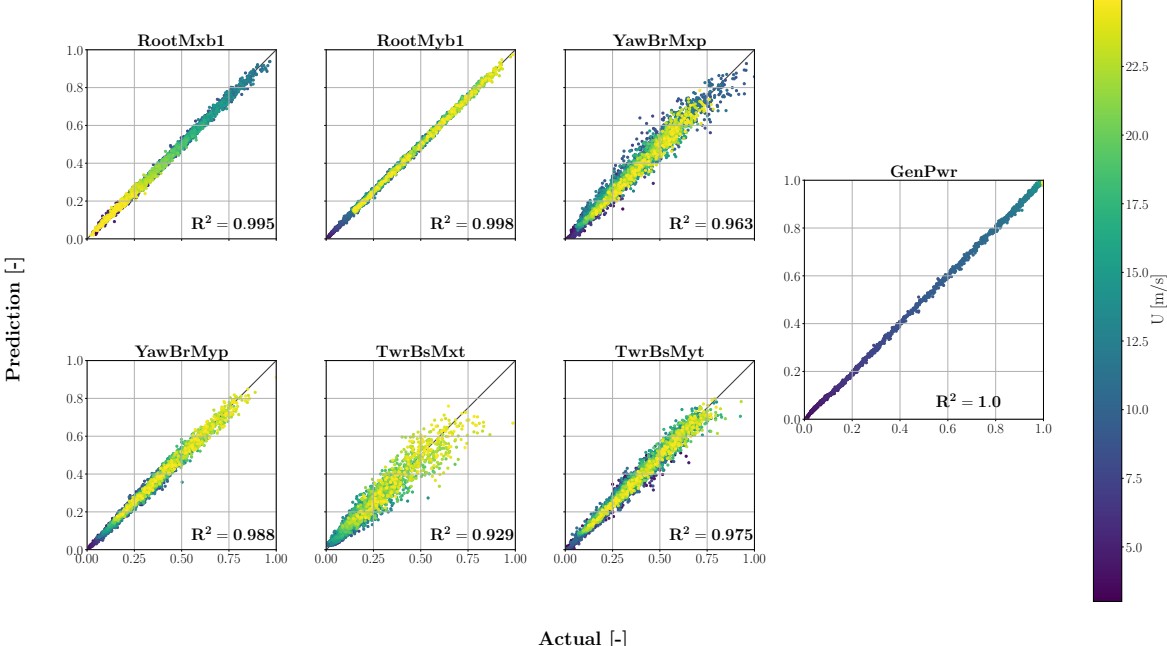

**Figure 9.** Testing results for TCN-FCNN trained SMs. The colour map represents a range of changes in the mean wind speed of the samples.

Although all channels have high $R^2$ values, the values decrease as we move from the blade root moments downwards. This decrease can be explained by the physical problem we are dealing with. The SMs have only one input, and they map wind input time series in $x$ direction to DEL outputs. The loads closer to the rotor are more affected by the wind input, while the structural dynamics of the wind turbine influence the further loads. Both fore-aft and side-side moments exhibit similar behaviour, but fore-aft moments are predicted more accurately. Therefore, it is reasonable to assert that the fore-aft moments are predominantly attributable to the wind, whereas the structural dynamics more significantly influence the side-to-side moments. To test this hypothesis, we infused the input of the TCN-FCNN input with the tower top side-side time series acceleration $a_{TTy}$. In other words, the TCN-FCNN maps the combination of synthetic wind time series *and* tower top side-side acceleration time





series to DEL. Figure 4 visually represents this network with tower top side-side acceleration as an optional input.

The modified SMs used for this are identical to those shown in Tables 2 and 5. The only difference is that the third dimension of the input shape in Table 5 has changed to 10 due to the concatenated time series. We followed the same process for training and testing these acceleration-enhanced SMs as we did for the original ones. In Figure 10, the data indicates that incorporating

the tower top acceleration time series into the input led to a better fit for the side-side moments, particularly for the tower bottom side-side moment. This confirms our hypothesis that these SMs can accurately capture the physics of the model at hand. One may argue that including the wind time series in the $y$ direction in the input would improve the tower bottom side-side moment $R^2$ value. We tested this hypothesis, but it did not improve the accuracy of the TCN-FCNN model.

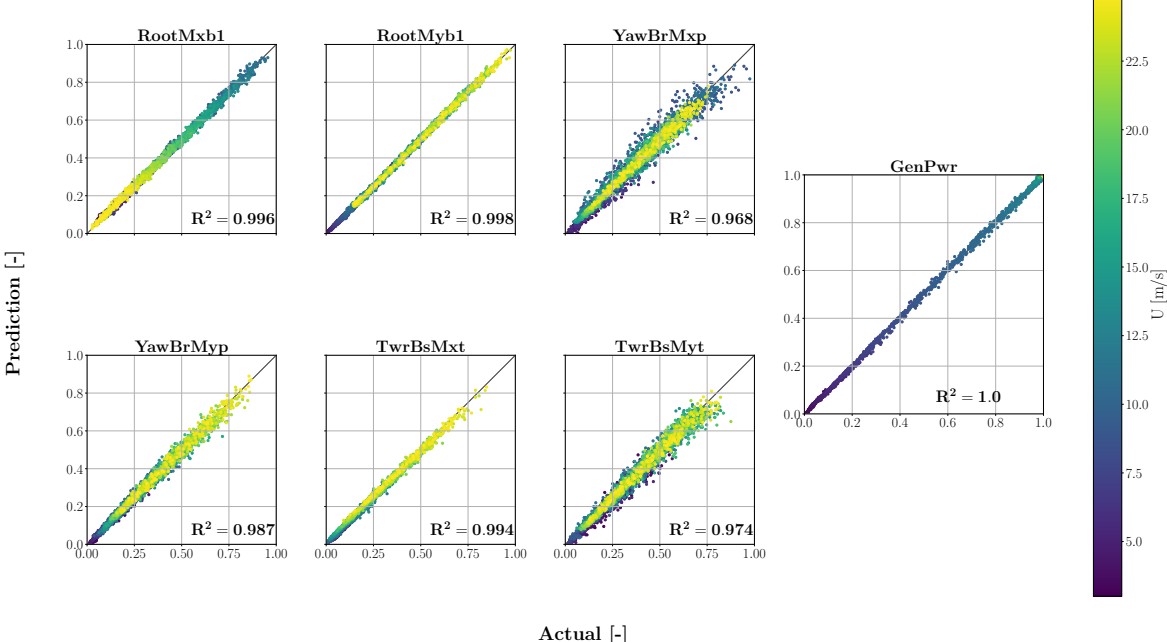

**Figure 10.** Testing results for TCN-FCNN trained SMs. The input for these SMs is infused with the tower top acceleration time series. The colour map represents a range of changes in the mean wind speed of the samples

## 470  3.6  FCNN results

As mentioned in Section 3.4, we trained and tested the FCNN SM on the three input variables samples, namely wind speed, wind shear and turbulence intensity. The FCNN aims to map the three input variables to DEL. This is very similar to the approach that was employed in Schröder et al. (2020). The results are presented in Figure 11. Similar to the TCN-FCNN





results, the $R^2$ values decrease from the top to the bottom of the turbine. Considering the simplicity of the FCNN SMs, they

perform very well in the testing phase.

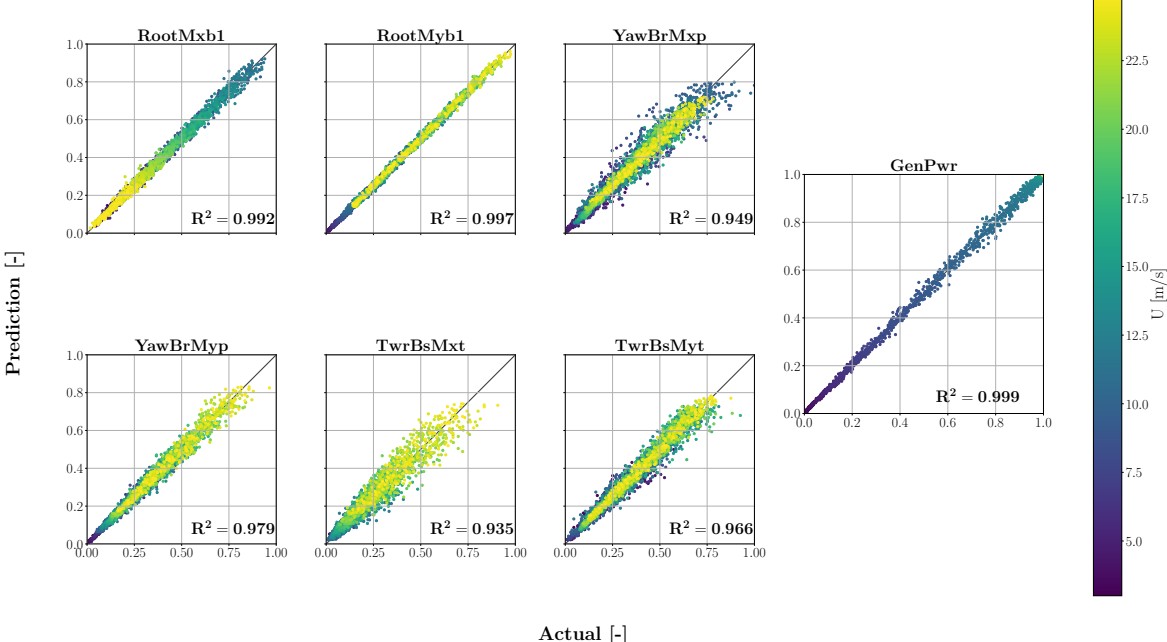

**Figure 11.** These are the testing results for the FCNN on the trained SMs.

## 3.7 Results comparison

To this point, we trained three different types of SMs on our data set. These SMs showed a good ability to predict the DEL from a limited amount of sequential data or input variable data. Each SM architecture has its own advantages and disadvantages. For TCN-FCNN architecture, the SMs can digest the complexities of wind time series. However, the model is complex and loses

accuracy for the channels that are not close to the rotor. On the other hand, FCNN architecture is simple and cheap to train, while it has higher $R^2$ values for the channels below the rotor. However, this model suffers from the same challenge in terms of $R^2$ values decline, and it is not prepared to take time series and needs input variables that may not be available all the time. To make the comparison more straightforward, the $R^2$ value for power and output channels and the required Graphics Processing Unit (GPU) time for the training is provided in Table 6.

The data presented in Table 6 reveals that all three SMs have consistently produced high $R^2$ values and low NRMSE. Notably, the utilization of tower top acceleration as the added input to wind has significantly enhanced the accuracy of prediction for the tower bottom side-side moment. This improvement is a testament to the TCN-FCNN SMs' ability to comprehend the mechanics behind the input-output correlation. Regarding the computational cost, FCNN SMs are more efficient for all the





**Table 6.** The accuracy of the fit and computational time for the SMs training. NRMSE is in percentage.

| Channel | TCN-FCNN | | | TCN-FCNN Accl. infused | | | FCNN | | |
| --- | --- | --- | --- | --- | --- | --- | --- | --- | --- |
| | $R^2$ | NRMSE | GPU Time [Sec] | $R^2$ | NRMSE | GPU Time [Sec] | $R^2$ | NRMSE | GPU Time [Sec] |
| GenPwr | 1.000 | 0.006 | 442 | 1.000 | 0.005 | 337 | 0.999 | 0.010 | 271 |
| RootMxb1 | 0.995 | 0.018 | 578 | 0.996 | 0.016 | 594 | 0.990 | 0.031 | 379 |
| RootMyb1 | 0.998 | 0.009 | 805 | 0.998 | 0.010 | 689 | 0.998 | 0.013 | 397 |
| YawBrMxp | 0.968 | 0.140 | 800 | 0.972 | 0.126 | 1042 | 0.987 | 0.190 | 110 |
| YawBrMyp | 0.987 | 0.039 | 807 | 0.986 | 0.040 | 781 | 0.996 | 0.064 | 397 |
| TwrBsMxt | 0.936 | 0.153 | 363 | 0.994 | 0.013 | 274 | 0.944 | 0.156 | 106 |
| TwrBsMyt | 0.971 | 0.094 | 613 | 0.977 | 0.099 | 756 | 0.977 | 0.119 | 163 |

outputs.

When examining Table 6, one might question the purpose of developing the TCN-FCNN SMs as they are more complex and computationally expensive. The TCN-FCNN approach offers a significant benefit by examining the wind's time series rather than solely its statistical properties. The DEL results from wind and/or wave time series oscillations. If we were to reduce these oscillations solely to wind or wave input statistics, this would undermine the accuracy of the DEL prediction. However, the TCN-FCNN can incorporate these oscillations and map them to the DEL. One challenge here is to free the input from the time series' length, which is not within the scope of this study. We will explore this further in our future studies. We briefly discuss the simulation length effect in Section 3.10.

We underline that the TCN-FCNN model can effectively decompose the wind field into its constituent features, which include the input variables. This capability was tested by expanding the feature vector in Figure 4 with the three input variables. Even with this augmentation, the $R^2$ values remained consistent, reaffirming the robustness of the TCN-FCNN in characterizing the wind field. Essentially, a latent space has been identified by the TCN suitable for accurate DEL prediction by the FCNN stage.

We tested the TCN-FCNN architecture to assess its ability to handle ultimate loads. During our analysis, we found that the SMs could accurately predict ultimate loads with a comparable level of precision as DEL prediction.

### 3.8 TCN-FCNN SMs in wake with TL

Considering the methodology employed for incorporating wake effects into our simulations, the use of a FCNN proves ineffective in this context. The FCNN relies on input parameters such as mean wind speed, turbulence intensity $TI$, and wind shear $\alpha$, which cannot adequately capture the complexities of wake interactions, as they cannot be condensed into a single scalar value. In a study by Dimitrov, an FCNN-based Surrogate Model was utilized to model wake effects. It was noted that their





**Table 7.** The results for the turbine in wake SMs in predictions after going through two stages of TL. The TL is done on both TCN-FCNN and the acceleration infused TCN-FCNN. For the sake of space acceleration infused TCN-FCNN indicates as TCN-FCNN A.I. NRMSE is in percentage.

| Wake centre | (-30m,90m) | | | | (0m,90m) | | | | (30m,90m) | | | |
| --- | --- | --- | --- | --- | --- | --- | --- | --- | --- | --- | --- | --- |
| | TCN-FCNN | | TCN-FCNN A.I. | | TCN-FCNN | | TCN-FCNN A.I. | | TCN-FCNN | | TCN-FCNN A.I. | |
| Channel | $R^2$ | NRMSE | $R^2$ | NRMSE | $R^2$ | NRMSE | $R^2$ | NRMSE | $R^2$ | NRMSE | $R^2$ | NRMSE |
| GenPwr | 0.999 | 0.998 | 0.999 | 1.214 | 0.998 | 1.949 | 0.999 | 1.002 | 0.999 | 1.275 | 0.999 | 0.996 |
| RootMxb1 | 0.970 | 3.546 | 0.972 | 3.654 | 0.961 | 4.381 | 0.918 | 5.281 | 0.969 | 3.689 | 0.971 | 3.642 |
| RootMyb1 | 0.979 | 3.335 | 0.972 | 3.580 | 0.975 | 3.793 | 0.962 | 4.844 | 0.968 | 4.234 | 0.976 | 4.636 |
| YawBrMxp | 0.952 | 4.787 | 0.953 | 4.617 | 0.960 | 3.808 | 0.938 | 5.756 | 0.960 | 4.560 | 0.955 | 4.690 |
| YawBrMyp | 0.925 | 6.783 | 0.960 | 5.104 | 0.923 | 7.593 | 0.962 | 4.899 | 0.909 | 7.957 | 0.950 | 5.540 |
| TwrBsMxt | 0.804 | 10.750 | 0.994 | 1.488 | 0.814 | 10.466 | 0.992 | 1.982 | 0.804 | 10.871 | 0.992 | 1.573 |
| TwrBsMyt | 0.858 | 7.837 | 0.944 | 4.312 | 0.911 | 5.643 | 0.944 | 5.310 | 0.939 | 5.343 | 0.955 | 4.927 |

model required additional inputs depending on the wind farm layout. In contrast, the TCN-FCNN approach, which relies solely on the flow information at the turbine location, demonstrates the capability to address wake challenges without necessitating
additional inputs, provided that the flow characteristics over the turbine are well-defined.

After training the TCN-FCNN and FCNN SMs, we tested the SMs on the input with the wake. The initial results without any TL did not provide an accurate prediction. Therefore, we used the TL as explained in Sections 2.5 and 3.3. Table 5 details the training setting for the TL models. The training is done on the $90\%$ of the turbine in wake databases, and testing is based
on the remaining $10\%$. We used both normal TCN-FCNN and acceleration infused TCN-FCNN in the TL for the turbine in the wake. The results for three wake centers are presented in Table 7.

The results predicted in Table 7 follow a similar trend to those obtained from the free stream turbine for all three wake centers. However, the SMs infused with acceleration provide higher $R^2$ values for almost all cases compared to the free stream, where the effect was mainly limited to tower bottom channels. Since the turbine's behaviour is more complex in the wake,
knowledge of its structural dynamics can be more influential in prediction. Therefore, including a channel from the turbine structure can aid the SMs in training a more accurate model and providing better predictions. It is worth noting that this training is conducted on only $1844$ out of $2048$ data points in the wake database, which is relatively small. Despite this, the model's ability to have a low NRMSE and high $R^2$ values demonstrates its strength.



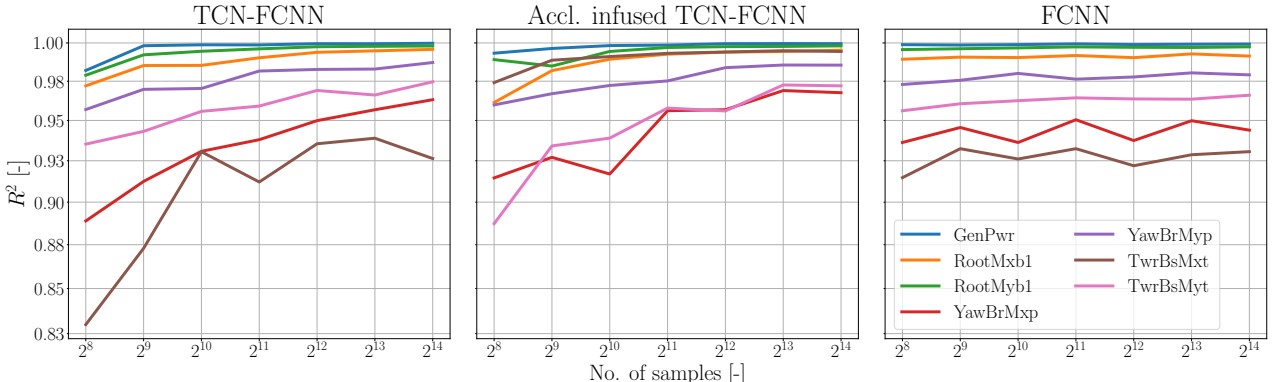

**Figure 12.** Sensitivity of $R^2$ to the number of training samples. The y-axis is on a logarithmic scale with a base of ten, while the x-axis is on a logarithmic scale with a base of two.

### 3.9 How much data is enough data?

One question that needs to be answered is how much data is enough to train these SMs accurately. In other words, we need to determine if reducing the number of sample points will affect the accuracy of the predictions made by the SMs. As we used the QMC method for sampling in this study, we can easily decrease the number of samples without having to redo the simulations. However, as our samples are based on Sobol's samples, we need to stick to the $2^n$ rule. To determine the amount of data needed, we trained the SMs on a smaller number of samples ranging from $2^6$ to $2^{14}$. Then, we randomly selected 1000 samples from the remaining input samples for prediction. For example, in the case of $2^{10}$, we trained the SMs on 1024 Sobol's samples data points and randomly selected 1000 samples from the remaining $2^{15} - 2^{10}$ samples for prediction. This enabled us to ensure fairness in comparing the $R^2$ values without any data leakage. The sensitivity analysis results in Figure 12 indicate that the $R^2$ value remains relatively high across all channels until the number of samples is reduced to $2^8$.

This shows that the SMs are versatile and do not require many sample points to make accurate predictions. The versatility of the SMs can be attributed to the simplicity and power of the models for providing accurate predictions and the effectiveness of Sobol's sampling in covering the input variable domains even with a low number of data points. This coverage helps the SMs interpolate well between the data they are not trained on. The displayed data in Figure 12 indicates that FCNN SMs exhibit less sensitivity to the number of samples. This observation aligns with the expectation that FCNN SMs possess a simpler architecture and fewer parameters to be trained. Therefore, the trained model improvement is minimal after passing the threshold of the number of samples. In the TCN-FCNN SMs, the improvement of the $R^2$ value varies depending on the channel. The tower channels exhibit a greater rate of $R^2$ improvement as compared to the rotor channels, with the side-side moment channels of the tower being the most prominent example. Additionally, acceleration infused TCN-FCNN shows higher improvement rates for tower top side-side and tower bottom fore-aft channels.





## 3.10  Time series length and data augmentation

In this study, we used a ten-minute time series for both input and output of the OpenFAST simulation. This is a common practice in the wind turbine engineering field as recommended by standards IEC 61400-1. Yet from the perspective of LiDAR and wind turbine controller, a ten-minute time series may be relatively long. Therefore, we aimed to include the ability to handle shorter synthetic wind time series and map them to DEL, which would be more attractive. Besides, in the field of machine learning, having enough training data is a challenge, and data augmentation is a solution (Mikołajczyk and Grochowski, 2018). To satisfy these two purposes, we attempted to augment data by dividing the 600-second synthetic wind and load time series into shorter segments and calculating DEL from those segments. If possible, the goal was to use even less than the minimum required number of simulations. However, our investigations have revealed that subsampling the 600-second load time series into segments shorter than 300 seconds adversely affects the accuracy of the DEL calculation. This is due to missed cycle counting of a the shorter data length obtained from OpenFAST output. Hence, this approach was not included in this manuscript.

## 4  Conclusion

This study explores the potential of employing a sequential ML model to develop a SM that correlates high-resolution wind time series with the DEL of wind turbine components. The methodology utilized in this manuscript involves creating a TCN-FCNN architecture for mapping synthetic wind time series to DEL, alongside a simpler FCNN for comparative purposes. We divided our methodology into twelve stages, including specifying the input variable space, generating synthetic wind time series, conducting aero-servo-elastic simulations, calculating DEL, splitting the data into training and testing databases, and building SMs. Also, we build a database of synthetic wind time series and DEL for a turbine in wake to test the versatility of the TCN-FCNN SMs using TL.

Our work begins with defining the input variable space and determining their boundaries, distributions, and sampling methods. We use a QMC Sobol's sampling technique to generate non-repetitive samples, guaranteeing uniformity, traceability and reproducibility. Next, we continued with the generation of synthetic wind time series using TurbSim, based on the input variable samples. These time series are stored in the Wind Database, forming the basis for subsequent simulations. The aero-servo-elastic simulations are performed using an NREL 5MW reference wind turbine model and OpenFAST, following IEC standards for power production DLC 1.2 (IEC 61400-1). The simulation output is stored in the Simulation Database, providing load time series data for various wind turbine components. With the loads time series data at hand, we calculate the DEL for each wind turbine component, adhering to the Palmgren-Miner linear damage rule. To train and test the SMs, we split the DEL database into training and testing sets, while ensuring no overlap between them. Two SM architectures were developed: a simple FCNN and a more complex TCN-FCNN. Both models are trained and tested to predict DEL based on unseen input variables or synthetic wind time series data. The FCNN SM serves as a benchmark for comparison with the more advanced





TCN-FCNN SM. By comparing the accuracy and performance of these models, we gain insights into the effectiveness of our approach. Moreover, we introduced the concept of testing our SMs in the context of a wake scenario. We created a new dataset that considers the wake effect on a wind turbine by implementing simplified wake models, thus expanding the versatility of our SMs.

In the results section, three different SM architectures were investigated: TCN-FCNN, FCNN, and an enhanced version of TCN-FCNN infused with tower top acceleration time series. The TCN-FCNN architecture was designed to take advantage of the wind time series data, making it capable of capturing the complex temporal dependencies in the wind field. Yet, its performance varied for different output channels, with higher $R^2$ values obtained for loads closer to the rotor and decreasing accuracy for loads influenced by structural dynamics. The addition of tower top acceleration time series as an input feature improved the accuracy of predicting the tower bottom side-side moment, demonstrating the SM's ability to discern relevant physics. In contrast, the FCNN architecture, which solely relies on input variables, offered a simpler and more efficient model with competitive predictive accuracy. The FCNN SMs performed well for all output channels, with $R^2$ values having an inverse relationship with the distance from the rotor. We further analyzed the trained TCN-FCNN models to determine how well they can predict DEL for a turbine situated in the downstream wake, a use case the FCNN SM could not tackle and illustrative of the TCN-FCNN architecture motivation. Our findings indicate that by using TL on a limited dataset, we can accurately forecast the DEL of a turbine in a wake. A sensitivity analysis was conducted to determine the minimum required sample size for training the SMs. It was found that both TCN-FCNN and FCNN SMs remained accurate with a relatively small number of samples, making them versatile and efficient for practical applications such as wind farm layout optimization. The choice between TCN-FCNN and FCNN SMs depends on the specific application requirements. TCN-FCNN is suitable when capturing fine-grained temporal dependencies in the wind field is crucial, while FCNN offers a simpler and more computationally efficient alternative with competitive performance. These SMs provide valuable tools for predicting DEL and enhancing wind turbine reliability, reducing the need for extensive and expensive sets of simulations. One of the drawbacks of this work is the input time signal length. One needs to investigate the possibility of liberating the SM from this time constraint, as it would make the model more versatile and more applicable to the wind speed time series of any length.

## 4.1 Future work

This study is the initial phase of building a ready-to-use and generalizable SMs for wind turbines. In this manuscript, we explore the ML based SM for this purpose and specifically TCN application in wind turbine engineering, while there is an extensive scope for further investigation. In future studies, we will implement TCN-FCNN methodology on an offshore wind turbine introducing complicated wave loading magnitude and directional spectra. Also, we will investigate the possibility of extending TL in a wind farm to train the SMs on one turbine and use transfer learning to build SMs for others in different wake conditions quickly. In this study, we took nine wind time series from the synthetic wind field as the input; therefore, reducing the number of wind time series is another interesting investigation alongside optimization of the placement of the points as a





hyperparameter.

Unfortunately, we were unable to access high-resolution wind turbine measurement data for our study. However, we recognize that incorporating this type of data into our methods could greatly enhance our research and should be a focus for future work. Additionally, we acknowledge that the synthetic data and mathematical models used in our SMs may not be as accurate as reality. As the saying goes, "All models are wrong, but some are useful." While the models that we use to build the databases and train the SMs may not be perfect, they still hold value. Therefore, one idea for future work is that the trained SMs can be

applied effectively on high-resolution measurement data by utilizing TL and inserting them between two trainable layers at the input and output stages. This approach can prove especially helpful when faced with limited measurement data.

*Code and data availability.*  The code supporting this study's findings is available from the corresponding author, RH, GitHub page. The data is available upon request.

*Author contributions.*  RH developed the necessary computer code and wrote the paper in consultation with and under the supervision of CC.

*Competing interests.*  The authors confirm there are no competing interests present.

*Acknowledgements.*  We greatly acknowledge the funding for this study by the Natural Sciences and Engineering Research Council of Canada (NSERC). This research was partly enabled by support provided by the Digital Research Alliance of Canada (alliancecan.ca).



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
