# Peer review of "Data-driven surrogate model for wind turbine damage equivalent load"

_Wind Energy Science, 2023_

## Referee Comment (RC1)

**Review of 'Data-driven surrogate model for wind turbine damage equivalent load', R. Haghi and C. Crawford**

This manuscript presents a neural network-based surrogate model of Damage Equivalent Loads (DEL) from aero-servo-elastic simulations of a wind turbine. It specifically generates synthetic wind timeseries from Sobol samples of inflow conditions (mean wind speed, turbulence intensity and wind shear) and utilizes a Temporal Convolutional Network (TCN) to project these into a latent space, with a Fully-Connected Neural Network (FCNN) topology acting as the read-out. In the results, the paper begins by i. benchmarking the TCN-FCNN model against a generic FCNN; ii. addressing worse side-to-side (SS) performance by infusing tower SS accelerations into the inputs, iii. demonstrating the TCN-FCNN expressiveness by estimating DELs under wake utilizing transfer learning and iv. discussing the minimal amount of data required.

The manuscript is well-written and the methodology well-described. The authors have produced an adequate amount of analysis. Moreover, the topic of surrogate models of numeric simulations is of great relevance for the wind community, and this work is a welcomed contribution.

There are, nonetheless, some comments that ought to be addressed in order to improve the overall quality of the paper.

Overall Methodology:

- There are good reasons for directly estimating DELs instead of bending moment reconstruction, but these are never provided in the manuscript. Moreover, it is explicitly stated that not only fatigue loads, but also maximal loads are of interest. Doesn't it make more sense then to reconstruct the bending moment signal and from it, extract the maximal loads and cycle count to get the DEL?
- In lines 65—80 there's a decent literature review of surrogate models of wind energy systems using neural networks. However, some important bibliographic elements are missing. Please consider reading and adding:
    o i. Movsessian A, Schedat M, Faber T. Feature selection techniques for modelling tower fatigue loads of a wind turbine with neural networks. Wind Energy Science 2021; 6(2): 539–554.
    o ii. d N Santos F, D'Antuono P, Robbelein K, Noppe N, Weijtjens W, Devriendt C. Long-term fatigue estimation on offshore wind turbines interface loads through loss function physics-guided learning of neural networks. Renewable Energy 2023
    o iii. Mylonas C, Abdallah I, Chatzi E. Deep unsupervised learning for condition monitoring and prediction of high dimensional data with application on windfarm scada data. In: Springer. 2020 (pp. 189–196).
    o iv. Vera-Tudela L, Kühn M. Analysing wind turbine fatigue load prediction: The impact of wind farm flow conditions. Renewable Energy 2017; 107: 352–360.
    o v. Duthé, G., de Nolasco Santos, F., Abdallah, I., Réthore, P.-É., Weijtjens, W., Chatzi, E., and Devriendt, C. (2023b). Local flow and loads estimation on wake-affected wind turbines using graph neural networks and pywake. In Journal of Physics: Conference Series, volume 2505, page 012014. IOP Publishing.

- - o vi. Liew, J., Riva, R., and Göçmen, T. (2023a). Efficient mann turbulence generation for offshore wind farms with applications in fatigue load surrogate modelling. In Journal of Physics: Conference Series, volume 2626, page 012050. IOP Publishing.

- Lines 84-86: 'The available literature and research indicate a lack of sufficient exploration and demonstration of a SM capable of mapping high-resolution environmental time series, specifically wind and/or wave for both on- and off-shore wind turbines, to the fatigue and extreme loads on wind turbine components.' I agree with the sentiment, our field requires more work on load surrogating of aeroelastic codes. However, I wouldn't say that there's a lack of exploration of SMs. There are many SM papers showing this can work (as pointed to in the previous comment), but we still need surrogates which are flexible enough (so it's positive that in the authors approach only u,ti and alpha are truly used as inputs) and access sufficient high quality data, which brings me to the assertion that you are using high-resolution environmental time series. Aerodynamically speaking, something like LES is more accurate than FAST. However, the major issue with any numeric simulator is two-fold: i. wave dynamics (for offshore) are really difficult to capture, ii. the controller is almost more of a black-box than ML. This nuance needs to be added into the discussion and limitations of any simulation-driven approach. I would also focus more on the innovativeness of using temporal convolutional networks.

- Concern with the train-test data split: a convolutional network which takes into account time-dependency (TCN) is being used. Even though DEL is estimated on a 10-min basis, given the time dependency I would argue you cannot simply randomly pick 90% of the data, it needs to be sequential (so, the 9 first months of a 10 month period).

- Table two as a reference to transfer learning (TL-FCNN). However, this isn't defined previously before showing up in this table.

- Section 3.1,.2,.3 and .4 aren't part of the results. I would suggest a new section dedicated to data generation including now section 3.1, 3.2 and 3.3. Section 3.4. is also more part of the methodology than the actual results.

- In figures 7b and 7c flap-wise and edge-wise aren't correct. It should be the other way around. Check figures of, e.g. Tartibu, L. K., M. Kilfoil, and A. J. Van Der Merwe. "Vibration analysis of a variable length blade wind turbine." (2012).

- Line 434. I don't understand the presence of SGD here. Above you mention you're using Adam (when you present the FCNN's topology).

- In Section 3.5. you present your results in terms of errors and predictions vs. true values. However, it would be important to actually plot the load signals (timeseries, vs wind speed, …) to understand the relative importance of each predicted quantity and their physical behaviour.

- Lines 466-467: 'One may argue that including the wind time series in the $y$ direction in the input would improve the tower bottom side-side moment $R2$ value. We tested this hypothesis, but it did not improve the accuracy of the TCN-FCNN model'. This is appreciated, but any claim you make needs to be substantiated. I.e., if you say you tested this hypothesis, then you need to show the results.

- You say in line 495 and around 'The TCN-FCNN approach offers a significant benefit by examining the wind's time series rather than solely its statistical properties. The DEL results from wind and/or wave time series oscillations. If we were to reduce these oscillations solely to wind or wave input statistics, this would undermine the accuracy of the DEL prediction.' You mentioned that the TCN can handle the complexities of wind timeseries over statistics and that this undermines accuracy, but the results don't show this. FCNN and TCN-FCNN results are similar. If you say that just wave and wind statistics are worse w.r.t. to timeseries

(intuitively, makes sense), then you need to prove this. Either you point to a reference showing this or you present results. However, if you see some of the suggested literature there are some fine examples that make it work.

- Lines 506-507: 'We tested the TCN-FCNN architecture to assess its ability to handle ultimate loads. During our analysis, we found that the SMs could accurately predict ultimate loads with a comparable level of precision as DEL.' You are again affirming something without proving it. You need to show results to make such an assertion.

- Lines 513-515: 'In contrast, the TCN-FCNN approach, which relies solely on the flow information at the turbine location, demonstrates the capability to address wake challenges without necessitating additional inputs, provided that the flow characteristics over the turbine are well-defined.' I wouldn't make this assertion. Wake, specially over an entire wind farm, isn't easily captured in terms of its loads by models based on a single turbine. There are very complex phenomena like boundary layer recuperation and wake effect accumulation which make it strongly non-linear.

- Section 3.10. There is a big problem with synthetically reducing your timeseries by cutting them: as you correctly point out, you'll miss a lot of cycles. Because you're applying an exponent (m), any missed long cycles (which are the ones you miss when you shorten the signal) will make your error explode because long cycles contribute very heavily for fatigue (even more so if your m is greater, 5, e.g.). Additionally, if you want to compare to industry, then the 10-minute window is pretty much standard.

- Conclusions: what is, in your opinion, the advantages of your timeseries TCN-FCNN approach over a statistics FCNN approach? How are these related to the operation of the turbine (e.g., rotor stops) and how to model these? What is in your opinion more important: a better model like TCN vs FCNN or including other data sources (acceleration)?

Clarifications:

- In lines 17-18 'The time-marching simulations are necessary for our work and research as they enable us to consider the inherent and necessary non-linearity in the wind turbine models', can you additionally explain how aeroelastic simulations encode non-linearities?

- Lines 50--. A quite thorough review of DTs for wind has been done but, after reading this section, there is no explicit mention to what in fact is a DT. Digital Twins are an often convoluted and overused concept, so it would be important for the authors to clearly state what they understand by digital twin and why it is different from a SM.

- From Section 2 it is not clear if the FCNN and TCN-FCNN training and testing samples are the same. They should be.

- How did you arrive to this topology present in Table 1? Was any hiperparameter tuning performed, any ablation studies? It is more correct to use search the variable space (randoms earch, Bayesian, etc.) in an automated manner, e.g. using Optuna or keras tuner.

- In lines 205--: 'CNNs have been used and are well known for classification proposes (Long et al., 2015). CNNs basics are well studied in the literature, and the interested reader is referred to Goodfellow et al. (2016); Long et al. (2015). Research has shown that TCN is better than Recurrent Neural Network (RNN) and LSTM in terms of performance, implementation, flexibility and versatility (Fawaz et al., 2019; Bai et al., 2018).' – When you refer that TCN are better than LSTMs and RNNs, is this also in the context of classification problems? It should be clear for the reader that you are using convolutional networks for regression.

- Line 209: 'a) the length of the output and input is the same'. How do you ensure (a) - length(input)=length(output) - if you have a timeseries of 10-minutes, but you only have a single 10-min value for DEL?

- Figure 3a and 3b. From both figures, it appears to me that the dilation factor already serves as a sort of dropout, or am I interpreting it incorrectly?

- Figure 4. Average pooling isn't defined elsewhere in the text.

- How did you arrive to the topology in Table 2? Also, if you're learning in the latent space, what is usually done (e.g. with an autoencoder) is to then have a read-out where the number of neurons per layer increases, e.g. 8,16,32. How were the number of neurons of the presented hidden layers selected?

- Line 293-294: '*neq* is the equivalent number of load cycles which is usually the length of the simulation in *s*.' By writing that neq is usually the length of the simulation it induces the reader to believe that neq is variable. Neq is a fixed number we use (almsot invariable 10e7, or lifetime DEL, or 1Hz DEL [which the authors use]) that enables us to compare different dynamic load timeseries by introducing the concept of equivalent load. This must be a constant throughout any period you are comparing (like the Wöhler exponent, it must remain constant). I don't understand what is meant here with neq = s. Additionally, what is the resolution of the DELs? 10-mins? This becomes clearer in subsequent sections, but it should be clearly stated when you introduce DELs that you're going to calculate them for a 10-minute time window.

- In Equation (6), why isn't the mean wind speed also dependent on time?

- Lines 373-374: 'For training and testing purposes, we only took into account nine synthetic wind time series in *x* direction out of 225 synthetic wind time series.' Does this mean that only 9 timeseries were used for training/testing or that only 9 points in the rotor plane were selected?

- Lines 431-432: 'Rather than training the SMs on all the training data, the training data set is divided into batches of 256 samples.' This sentence induces the read into a wrong idea. Batch training is 'training on all the training data'. The model still see the full dataset set for each epoch, just divided into batches.

- In lines 465-466 you notice how the accelerations improve the performance, specifically for the tower bottom. It is however interesting how the greatest improvement is at the bottom and not the top, where you have the sensor installed. Could you perhaps expand on this, why does it happen and specifically the relation (or relative lack thereof) between tower bottom's bending moment and the structural dynamics at the rotor level.

- Line 482. You say that FCNN 'needs input variables that may not be available all the time'. But this critique can also be made of TCN-FCNN and even more so: the probability of models based on 1Hz data failing is greater than on 10-min statistics.

- Line 496: 'One challenge here is to free the input from the timeseries' length, which is not within the scope of this study.' What is meant by this?

Technical corrections (typos, grammar, etc.):

- In the abstract, the sentence 'Doing so can calculate fatigue and extreme loads on the wind turbine's components' is convoluted and doesn't seem to be clear English.
- Line 65. Remove character 'f'.
- Line 292 – 'Wöhler slope'. Wöhler exponent or inverse of the S-N curve slope.
- The correct SI unit for second is s, not sec.
- Line 510 'wind speed'. Add mean(u).
- Line 512. Missing citation in Dimitrov.

Suggestions for better readability and more complete content (implementation left at your own good judgement):

- In Figure 1, suggestion to change the order: top data generation, bottom SM. We read from top to bottom and sequentially, it makes more sense to present first data generation, followed by SMs. Also, I suggest to clearly label the FCNN as the 'baseline' or 'benchmark' and alter its color w.r.t. TCN-FCNN.
- The inputs used are mean wind speed, turbulence intensity and wind shear. However, in the real world there is usually a degree of yaw misalignment. You can maybe include this randomly into your input variable space (yaw misalignment).
- Figure 5 could be clearer. Why is fine-tune step trainable also include the TCN? Isn't it clearer to simply point the frozen weights of TCN and then say that FCNN can be re-trained?
- Perhaps you might consider publishing your simulation datasets under an open access license, e.g. in Zenodo.
- In Figure 8 you're taking snapshots, but maybe a better way to compare with the Gaussian velocity deficit is to time-average it.
- You trained 6 independent model. It would possibly reduce the performance (marginaly, one would expect), but you could think of training multi-objective NNs.

---

## Author Comment (AC1)

| | Comment | Response |
|---|---|---|
| 1 | Consider re-organizing the sections as there are currently methodology aspects on results. | The authors prefer to keep the paper organizations as they are. Regarding NREL 5MW characteristics, the authors cited NREL documents with all turbine details. There is not site in this study. We took samples from U, TI and alpha based on the defined distrubutins in the manuscript. As there are only two turbines, the turbine layout does not seem required in this case. |
| 2 | The actual response of the turbine should be included as results. This will help the readers see how are the fatigue loads distributed for the free-stream and wake operating turbines. Consider showing DEL vs mean wind speed. | We added a plot with the distribution of DEL for each output. |
| 3 | It is a well reported fact that a significant amount of the variability of the load response of the turbine is due to the different realizations of the turbulent flow fields. Note, that two turbulent fields can have the same flow parameters (U, TI, shear) but have different turbulent structures and therefore very different DEL's. It is very common on literature to use multiple seeds in the turbulent inflow generation in order to quantify this impact. The IEC 61400 recommends to use at least 6 seeds, but several of the references you cite report that a larger number is necessary, for example Liew et al. (DOI 10.1088/1742-6596/2265/3/032049) recommend 21 seeds. | It is worth checking for sure. However, as we are taking Sobol's samples, we already covered those different TIs for a wind speed. Also, we have hardware limitation and this number of simulations was the limit we could go. Even two seeds per sample would have been too much for our setup. |
| 4 | The authors should consider using the 3 components of turbulence instead of only longitudinal component, as the v, and w components have a noticeable impact on the response. Additionally, the authors should consider increasing the resolution of the turbulent fields as they are on the coarser end (15x15). | For running the simulaitons, we took into account all the wind components. However, for training and testing we did not see any improvement in the results by taking into account u, v and w from the wind field. The models are porvided and the interested reader can use them to test them. The increase in the number of grid points would increases the turbsim output file size while it would not have a meaninngful effect of on the presented methodology. Therefore, due to hardware limits we decided to go for a 15x15 grid. |
| 5 | The article does not provide any insights on how was the NN architecture was selected for both FCNN and TCN-FCNN. This should be part of the article. | We architecture obtained exprimentally. |
| 6 | TCN is used as a dimensional reduction or feature extraction step. Then a FCNN is used to map the latent variables to the DEL. Please report the number of latent variables used (shape of feature vector). It would be interesting to present the dependency of the DEL on the latent space variables (maybe for few examples). | The features vector length is added to Table 2. The sensitivity analysis is out of the scope of this work. |

| 7 | Why are only 9 of the 15x15 inflow field time-series are selected to be used with TCN-FCNN? One would expect that the dimensional reduction algorithm (TCN) should benefit from more data, and therefore extract better quality latent variables if feed with all the data. | The point for us was figuring out the minimum amount of data in windfield which is required to have an accurate prediction. The end goal is having a model that can predict the DEL only based on one wind time series. We tested this for one, to 9 time series, and we figure out the minimum number for an acceptable accuracy is 9 wind time series. Increasing the number of wind time series did not have any significant effect on the accuracy of the prediction and made the training training/testnig longer. Therefore, we decided to use only 9 time series from the wind field. |
|---|---|---|
| 8 | The application of the article is turbine load surrogates based on inflow parameters and/or inflow fields. This application makes sense for turbine design and site suitability. But later on the article a load surrogate that uses both inflow and tower top acceleration signals is introduced. These type of surrogates have a different application, such as virtual sensors or digital twin as the accelerations signals are not available on turbine design without performing an aeroelastic simulation. The article could benefit for a clearer statement of the intended applications of the different surrogates. Maybe you could consider dropping the virtual sensor application on this article, and maybe plan another article where you apply you surrogate techniques on setups that challenge the DT application for example testing the surrogates using lidar as inflow measurements, and SCADA data that contains accelerometers. | This is a valid point and authors agree with this. We the manuscript to include statements about DT and virtual sensing. |
| 9 | Consider adding a discussion about the impact of non-stationary wakes on the accuracy results of your TCN-FCNN SM. Currently you consider a stationary wake model for deficit and added turbulence, this means that the effect of the wakes on the inflow-time-series inputs to your SM consists on shifting the means and scaling the standard deviations, it is unclear to me that your SM would have the same accuracy if tested on higher fidelity models such as dynamic wake meandering or large eddy simulations, on cases where the dynamics of the inflow are altered. | This is a very interesting idea, and further exploration is required. This is added to the future work as DMW and high fidelity simulations is out of this work scope. |

---

## Author Comment (AC2)

| Comment | Response |
|---|---|
| There are good reasons for directly estimating DELs instead of bending moment reconstruction, but these are never provided in the manuscript. Moreover, it is explicitly stated that not only fatigue loads, but also maximal loads are of interest. Doesn't it make more sense then to reconstruct the bending moment signal and from it, extract the maximal loads and cycle count to get the DEL? | That is a valid comment. We tried to map the wind time series to the moments/forces time series, but we did not succeed. At the same time, usually in wind engineering practice, we only use the moments/forces time series to extract the ultimate and fatigue loads. Then we do not have that much use for them; therefore, if it is possible to skip a step, it would accelerate the whole process. |
| In lines 65–80 there's a decent literature review of surrogate models of wind energy systems using neural networks. However, some important bibliographic elements are missing. Please consider reading and adding: | Thanks for the list, part of the recommended literature are added to the introduction. |
| However, I wouldn't say that there's a lack of exploration of SMs. There are many SM papers showing this can work (as pointed to in the previous comment), but we still need surrogates which are flexible enough (so it's positive that in the authors approach only u, ti, and alpha are truly used as inputs) and access sufficient high-quality data, which brings me to the assertion that you are using high-resolution environmental time series. Aerodynamically speaking, something like LES is more accurate than FAST. However, the major issue with any numeric simulator is two-fold: i. wave dynamics (for offshore) are really difficult to capture, ii. the controller is almost more of a black-box than ML. This nuance needs to be added into the discussion and limitations of any simulation-driven approach. I would also focus more on the innovativeness of using temporal convolutional networks. | Thanks for your comments. We added some lines in the introduction and conclusion sections to address your argument. |
| Concern with the train-test data split: a convolutional network which takes into account time-dependency (TCN) is being used. Even though DEL is estimated on a 10-min basis, given the time dependency I would argue you cannot simply randomly pick 90% of the data, it needs to be sequential (so, the 9 first months of a 10 month period). | This is a lack of clarity from our side. We did not use 90% of the time series length for training and then 10% for testing. We utilized 90% of the 32726 time series each 600s and corresponding DELs for training, and use 10% for testing. This is clarified in the updated manuscript. Where the randomness comes to play is the selection of the 90% of the samples. |
| Table two as a reference to transfer learning (TL-FCNN). However, this isn't defined previously before showing up in this table. | Yes, this is to save space and not repeat the same table. I added a sentence in the caption to explain the TL-FCNN refers to section 3.4. |
| Section 3.1, 3.2, 3.3 and 3.4 aren't part of the results. I would suggest a new section dedicated to data generation including now section 3.1, 3.2 and 3.3. Section 3.4. is also more part of the methodology than the actual results. | I updated the paper structure accordingly. |
| In figures 7b and 7c flap-wise and edge-wise aren't correct. It should be the other way around. Check figures of, e.g. Tartibu, L. K., M. Kilfoil, and A. J. Van Der Merwe. "Vibration analysis of a variable length blade wind turbine." (2012). | Fixed in the updated version |
| Line 434. I don't understand the presence of SGD here. Above you mention you're using Adam (when you present the FCNN's topology). | Fixed in the updated version |
| In Section 3.5. you present your results in terms of errors and predictions vs. true values. However, it would be important to actually plot the load signals (time series, vs wind speed, ...) to understand the relative importance of each predicted quantity and their physical behavior. | We do not take into account any time series in our predictions. However, I added both free stream and wake raw DEL for six channel distributions to the paper. |

| | |
|---|---|
| Lines 466-467: 'One may argue that including the wind time series in the y direction in the input would improve the tower bottom side-side moment R2 value. We tested this hypothesis, but it did not improve the accuracy of the TCN-FCNN model'. This is appreciated, but any claim you make needs to be substantiated. I.e., if you say you tested this hypothesis, then you need to show the results. Lines 506-507: 'We tested the TCN-FCNN architecture to assess its ability to handle ultimate loads. During our analysis, we found that the SMs could accurately predict ultimate loads with a comparable level of precision as DEL.' You are again affirming something without proving it. You need to show results to make such an assertion. | For the sake of space, I will not include those results. However, I uploaded the trained model for DEL and ultimate loads with the databases on the Zolondo. Interested readers can download the data, and trained model to verify the statement. |
| You say in line 495 and around 'The TCN-FCNN approach offers a significant benefit by examining the wind's time series rather than solely its statistical properties. The DEL results from wind and/or wave time series oscillations. If we were to reduce these oscillations solely to wind or wave input statistics, this would undermine the accuracy of the DEL prediction.' You mentioned that the TCN can handle the complexities of wind time series over statistics and that this undermines accuracy, but the results don't show this. FCNN and TCN-FCNN results are similar. If you say that just wave and wind statistics are worse w.r.t. to time series (intuitively, makes sense), then you need to prove this. Either you point to a reference showing this or you present results. However, if you see some of the suggested literature there are some fine examples that make it work. | Thanks for the comment. I checked the examples and updated the statements accordingly. |
| Lines 513-515: 'In contrast, the TCN-FCNN approach, which relies solely on the flow information at the turbine location, demonstrates the capability to address wake challenges without necessitating additional inputs, provided that the flow characteristics over the turbine are well-defined.' I wouldn't make this assertion. Wake, especially over an entire wind farm, isn't easily captured in terms of its loads by models based on a single turbine. There are very complex phenomena like boundary layer recuperation and wake effect accumulation which make it strongly non-linear. | I agree with nonlinearity, but what I claim is about the versatility of TCN-FCNN, which can map the inflow of a turbine in wake to its DEL. I added a couple of sentences to explain this better. |
| Section 3.10. There is a big problem with synthetically reducing your time series by cutting them: as you correctly point out, you'll miss a lot of cycles. Because you're applying an exponent (m), any missed long cycles (which are the ones you miss when you shorten the signal) will make your error explode because long cycles contribute very heavily for fatigue (even more so if your m is greater, 5, e.g.). Additionally, if you want to compare to industry, then the 10-minute window is pretty much standard. | I added your comment to the text for clarification. |
| Conclusions: what is, in your opinion, the advantages of your time series TCN-FCNN approach over a statistics FCNN approach? How are these related to the operation of the turbine (e.g., rotor stops) and how to model these? What is in your opinion more important: a better model like TCN vs FCNN or including other data sources (acceleration)? | These questions are answered in the updated manuscript. |

| | |
|---|---|
| Lines 17-18 'The time-marching simulations are necessary for our work and research as they enable us to consider the inherent and necessary non-linearity in the wind turbine models', can you additionally explain how aeroelastic simulations encode non-linearities? | This is out of this manuscript's scope. However, I added some references that included some explanations. |
| Lines 50–. A quite thorough review of DTs for wind has been done but, after reading this section, there is no explicit mention to what in fact is a DT. Digital Twins are an often convoluted and overused concept, so it would be important to clearly state what in fact is a DT. | I agree that DT description is all over the board, and I am not interested in entering that discussion in this paper. However, I added a short statement to distinguish the DT from SM in the manuscript. |
| From Section 2 it is not clear if the FCNN and TCN-FCNN training and testing samples are the same. They should be. | They are not. I added a short statement about it to the manuscript. In short, as we take 90% of the data for training, and this is based on Sobol's samples they have a large overlap. Also, this randomness shows generalizability of the models. |
| How did you arrive at this topology present in Table 1? Was any hyperparameter tuning performed, any ablation studies? It is more correct to use search the variable space (random search, Bayesian, etc.) in an automated manner, e.g. using Optuna or keras tuner. | It is more accurate to use the Keras tuner, and discover the design space indeed. But, as this was a simple three-layer network, the architecture and hyperparameters were obtained experimentally. |
| Lines 205–: 'CNNs have been used and are well known for classification purposes (Long et al., 2015). CNNs basics are well studied in the literature, and the interested reader is referred to Goodfellow et al. (2016); Long et al. (2015). Research has shown that TCN is better than Recurrent Neural Network (RNN) and LSTM in terms of performance, implementation, flexibility and versatility (Fawaz et al., 2019; Bai et al., 2018).' – When you refer that TCN are better than LSTMs and RNNs, is this also in the context of classification problems? It should be clear for the reader that you are using convolutional networks for regression. | They also perform better than LSTM and RNN for classification according to the cited literature. However, for our purpose, we only talk about regression. This is clarified in the updated manuscript. |
| Line 209: 'a) the length of the output and input is the same'. How do you ensure (a) - length(input)=length(output) - if you have a time series of 10-minutes, but you only have a single 10-min value for DEL? | Here we only talk about TCN and not TCN-FCNN. The output of the TCN part of the proposed architecture is equal to the length of the input. Then, it goes through the FCNN part to turn into a zero-dimension array (DEL). |
| Figure 3a and 3b. From both figures, it appears to me that the dilation factor already serves as a sort of dropout, or am I interpreting it incorrectly? | Your interpretation is right. |
| Figure 4. Average pooling isn't defined elsewhere in the text. | I added a reference to Goodfellow's book for the interested reader to read about different layers in a NN architecture. |
| How did you arrive at the topology in Table 2? Also, if you're learning in the latent space, what is usually done (e.g. with an autoencoder) is to then have a read-out where the number of neurons per layer increases, e.g. 8,16,32. How were the number of neurons of the presented hidden layers selected? | The latent space here refers to a representation of the data in lower dimensions while it preserves the important qualities of the data. Therefore, it doesn't need to be an AE. The number of neurons and the FCNN architecture are obtained experimentally. It was not an efficient way, but it was the best the authors knew at the time of this project. |

| | |
|---|---|
| Line 293-294: 'neq is the equivalent number of load cycles which is usually the length of the simulation in s.' By writing that neq is usually the length of the simulation it induces the reader to believe that neq is variable. Neq is a fixed number we use (almost invariable 10e7, or lifetime DEL, or 1Hz DEL [which the authors use]) that enables us to compare different dynamic load timeseries by introducing the concept of equivalent load. This must be a constant throughout any period you are comparing (like the Wöhler exponent, it must remain constant). I don't understand what is meant here with neq = s. Additionally, what is the resolution of the DELs? 10-mins? This becomes clearer in subsequent sections, but it should be clearly stated when you introduce DELs that you're going to calculate them for a 10-minute time window. | Valid points. Neq here is the length because we are at 1Hz sampling. It is clarified in the updated manuscript. |
| In Equation (6), why isn't the mean wind speed also dependent on time? | TurbSim combines the mean wind speed profile over the rotor, which includes shear and veer depending on height, with turbulent fluctuations that have zero mean. The mean wind speed doesn't depend on time. A short explanation was added to the updated manuscript. |
| Lines 373-374: 'For training and testing purposes, we only took into account nine synthetic wind time series in x direction out of 225 synthetic wind time series.' Does this mean that only 9 timeseries were used for training/testing or that only 9 points in the rotor plane were selected? | The nine indicated points on the rotor time series are utilized for training and testing. It is clarified in the updated manuscript. |
| Lines 431-432: 'Rather than training the SMs on all the training data, the training data set is divided into batches of 256 samples.' This sentence induces the reader into a wrong idea. Batch training is 'training on all the training data'. The model still see the full dataset set for each epoch, just divided into batches. | Agreed. It is clarified in the updated manuscript. |
| In lines 465-466 you notice how the accelerations improve the performance, specifically for the tower bottom. It is however interesting how the greatest improvement is at the bottom and not the top, where you have the sensor installed. Could you perhaps expand on this, why does it happen and specifically the relation (or relative lack thereof) between tower bottom's bending moment and the structural dynamics at the rotor level. | The side-side tower top moment is mainly a result of the wind turbine rotor torque, and it is not affected by the tower top side-side acceleration. However, the tower's bottom side-side bending is caused by the side-side forces at the tower top, which in this case is represented by the side-side acceleration. It is explained in the updated manuscript. |
| Line 482. You say that FCNN 'needs input variables that may not be available all the time'. But this critique can also be made of TCN-FCNN and even more so: the probability of models based on 1Hz data failing is greater than on 10-min statistics. | This is a valid point. But, here, I am not necessarily talking about this model. This model is an example of a model that can map wind time series to DEL. Regarding the FCNN, in reality, the TI and wind shear are not necessarily available in what a SCADA collects. However, this methodology will hopefully work with wind speed time series measurements. This is clarified in the updated manuscript. |
| One challenge here is to free the input from the time series' length, which is not within the scope of this study.' What is meant by this? | So, in this manuscript, we trained a model to map 10-minute time series to DEL. Each of these 10-minute wind time series has a specific mean wind speed, shear, and TI. However, in reality, these things are changing in 10 minutes. Therefore, in the future, it is important to improve the model to be able to map the changing wind time series to DEL. |

---

## Author Response (AR1)

Dear Editor,

Thank you very much for accepting this manuscript for review. We have taken into account the majority of the reviewer's comments. A detailed response to the reviewer's comments can be found in the interactive section of the review process. However, some of the main changes are indicated here:

- The structure of the manuscript has been updated to have a separate section for data generation.

- The explanation about the testing-training datasets has been clarified.

- Figure 7 is corrected.

- We have expanded the introduction and previous work section as recommended by one of the reviewers.

- The motivation has been clarified, and the limitations of the work are extended.

We would like to thank the reviewers for their constructive comments that helped us improve the quality of the manuscript.

Kind regards,
Rad Haghi

---

## Author Response (AR2)

Dear Editor,

Thank you very much for accepting this manuscript for review and for your helpful comment on the minor revision. We added a statement to Section 2.7 of the manuscript to answer your comment.

Kind regards,
Rad Haghi